# Evaluation of Geospatial Interpolation Techniques for Enhancing Spatiotemporal Rainfall Distribution and Filling Data Gaps in Asir Region, Saudi Arabia

Ahmed M. Helmi [1,*], Mohamed Elgamal [2], Mohamed I. Farouk [2,3], Mohamed S. Abdelhamed [4] and Bakinam T. Essawy [5]

1 Irrigation and Hydraulics Department, Faculty of Engineering, Cairo University, Giza 12613, Egypt
2 Civil Engineering Department, College of Engineering, Imam Mohammad Ibn Saud Islamic University (IMSIU), Riyadh 13318, Saudi Arabia; mhelgamal@imamu.edu.sa (M.E.); miradi@imamu.edu.sa (M.I.F.)
3 Irrigation and Hydraulics Department, Faculty of Engineering, Ain Shams University, Cairo 11517, Egypt
4 Global Institute for Water Security, University of Saskatchewan, Saskatoon, SK S7N 3H5, Canada; mohamed.abdelhamed@usask.ca
5 Civil and Infrastructure Engineering and Management Department, Nile University, Sheikh Zayed City 12588, Egypt; btaessawy@nu.edu.eg
* Correspondence: ahmed.helmi@eng.cu.edu.eg

**Abstract:** Providing an accurate spatiotemporal distribution of rainfall and filling data gaps are pivotal for effective water resource management. This study focuses on the Asir region in the southwest of Saudi Arabia. Given the limited accuracy of satellite data in this arid/mountain-dominated study area, geospatial interpolation has emerged as a viable alternative approach for filling terrestrial records data gaps. Furthermore, the irregularity in rain gauge data and the yearly spatial variation in data gaps hinder the creation of a coherent distribution pattern. To address this, the Centered Root Mean Square Error (CRMSE) is employed as a criterion to select the most appropriate geospatial interpolation technique among 51 evaluated methods for maximum and total yearly precipitation data. This study produced gap-free maps of total and maximum yearly precipitation from 1966 to 2013. Beyond 2013, it is recommended to utilize ordinary Kriging with a J-Bessel semivariogram and simple Kriging with a K-Bessel semivariogram to estimate the spatial distribution of maximum and total yearly rainfall depth, respectively. Additionally, a proposed methodology for allocating additional rain gauges to improve the accuracy of rainfall spatial distribution is introduced based on a cross-validation error (CVE) assessment. Newly proposed gauges in the study area resulted in a significant 21% CVE reduction.

**Keywords:** geospatial interpolation; data gaps; kriging; optimum rainfall locations; arid region

## 1. Introduction

Arid lands cover 41% of the Earth's surface and sustain one-third of the global population, half the world's livestock, and around 44% of its food production [1]. These regions, characterized by low rainfall and high evaporation rates, have witnessed heightened water demand in recent decades, intensifying pressure on freshwater resources and amplifying water scarcity concerns [2,3]. The irregular occurrence of droughts and floods in these areas presents substantial challenges to water resource management [4]. Moreover, optimizing irrigation practices and enhancing crop yields, especially in water-scarce arid regions, relies on a comprehensive understanding of rainfall distribution [5]. Analyzing these patterns facilitates the identification of drought- and flood-prone zones, enabling proactive mitigation strategies. Predicting rainfall in arid regions poses challenges due to intricate interactions among meteorological factors such as air temperature, humidity, and wind, further compounded by the scarcity of comprehensive meteorological networks, which

hinders precise assessment efforts [6–9]. Navigating the sporadic and unpredictable nature of rainfall patterns necessitates the inclusion of additional environmental determinants, such as the topography and elevation of different locations [10,11]. Hence, accurate rainfall measurement and prediction are vital for effective water management in such strategic regions.

The emergence of spatial data and distributed models, coupled with advanced computing capabilities, has collectively led to the integration of these models into scientific research due to their ability to simulate intricate flow regimes within complex study domains [12–14]. Factors including spatial rain gauge distribution, elevation, and historical data length contribute to the precision of spatial rainfall estimation at targeted sites. These factors affect the simulation of various hydrological processes like runoff, recharge, soil moisture, and evaporation [4,15,16]. Therefore, ensuring accurate gridded rainfall data is crucial to meet the demands of these models.

Ground truth rainfall records provide valuable insights for calibration, identifying trends and patterns, analyzing the impact of climate changes, and correcting biases in current climatic models and their future projections [17,18]. The number of rain gauges required to obtain accurate rainfall estimates depends on the size of the area under consideration and the desired level of accuracy [6]. Inadequate gauge numbers can result in underestimation or overestimation of rainfall amounts, leading to erroneous conclusions about the water balance within the studied region [19,20]. Challenges are also apparent in data accessibility and limitations, such as brief availability or data gaps, while data reliability may be compromised due to measurement errors or inhomogeneities and potential unavailability [19]. Even with a well-distributed gauge network, some dysfunction during storm events exacerbates the challenges. One potential issue when examining rainfall data involves missing records due to various factors, including discontinued measurements, and damaged/displaced rainfall gauges following flood events, among others. These data gaps can yield false trends or spatial correlations that inaccurately represent natural variations [21,22]. Researchers employ advanced modeling techniques and satellite data to address these challenges and enhance the precision of rainfall prediction in arid regions. These techniques facilitate a comprehensive understanding of the factors influencing rainfall patterns and provide details about spatial distribution [12,14,20,21]. From a hydrological standpoint, assessing the reliability of existing ground truth rain gauges in estimating rainfall data in areas lacking gauge information is pivotal.

The ground rain gauge station network in arid regions exhibits a dual nature, characterized by coarse spacing adhering to World Meteorological Organization (WMO) guidelines for minimum density and historical rainfall data often containing gaps that need filling before use [23]. Several techniques are detailed in the literature to address these gaps, including using satellite data [24–27], implementing geostatistical interpolation techniques between terrestrial rain gauges to conduct regional rainfall analyses from available data measurements as detailed in Table 1, or using machine learning and statistical analysis [28]. Although satellite-based rainfall estimates can provide valuable information, their limitations in accuracy and spatial resolution emphasize the necessity of ground truth rain gauges and appropriate interpolation methods to improve the accuracy of rainfall estimates. For instance, satellite data have not yet provided a satisfying accuracy for total and maximum yearly precipitation information in the Arabian Peninsula, making geostatistical interpolation between terrestrial rain gauges crucial for obtaining reliable rainfall data [29]. Therefore, it is essential to have an adequate density of rain gauges over the area under consideration to obtain accurate rainfall estimates.

Spatial interpolation techniques are widely utilized to enhance the spatial resolution of data through the estimation of values in areas without sampling, applied in geosciences, water resources, environmental sciences, agriculture, and civil engineering [30,31]. However, their accuracy is influenced by factors such as sampling density, sample spatial distribution, sample clustering, surface type, data variance, data normality, quality of secondary information, stratification, and grid size or resolution [32,33], sometimes resulting

in underestimations or overestimations. To enhance accuracy, ground truth gauges are used for calibration in these techniques [18]. The selection of an interpolation method, whether deterministic or geostatistical, depends on the trade-off between accuracy and computational efficiency. Deterministic techniques are more straightforward and faster but do not account for the uncertainty associated with the estimation process [34], while geostatistical techniques provide a comprehensive view of the uncertainty associated with the estimation process but are computationally more intensive [35,36]. As a result, several spatial interpolation techniques have been developed that are appropriate for the rapid estimation process [20,37–41].

To evaluate the efficiency of spatial interpolation techniques, it is imperative to consider the following key factors: (1) the design of the sampling process; (2) the mean and coefficient of variation of the primary variable for either the estimation dataset or validation dataset; (3) the sample size for both the estimation and validation datasets; (4) the geographical extent of the study area; and (5) the use of appropriate accuracy measurements [31]. Studies have analyzed spatial interpolation techniques (see Table 1); however, comprehensive conclusions remain elusive. Despite this, the three most frequently compared techniques are Ordinary Kriging (OK), Inverse Distance Weighting (IDW) with inverse distance squared (IDS), while different types of Kriging geostatistical techniques were evaluated: Ordinary Kriging, Cokriging, and Empirical Bayesian Kriging.

**Table 1.** Sample of available comparative studies in the literature.

| Interpolation Method | Validation Method | Recommended Method | Ref. |
|---|---|---|---|
| Thiessen Polygon (TB), Inverse Distance Weighting (IDW), Linear Regression (LG), Kriging with External Drift (KED), Ordinary Kriging (OK) | Root Mean Squared Error (RMSE) | Ordinary Kriging (OK) | [42] |
| Inverse Distance Weighting (IDW), Local Polynomial Interpolation (LPI) Global Polynomial Interpolation (GPI) Simple Kriging (SK) Universal Kriging (UK), Ordinary Kriging (OK), Radial Basis Function (RBF) | Mean Error (ME) Root Mean Squared Error (RMSE) | Ordinary Kriging (OK) | [43] |
| Natural Neighbor Interpolation (NNI), Ordinary Kriging (OK) Cokriging (CK) | Root Mean Squared Error (RMSE) | Cokriging (CK) | [44] |
| Kriging with External Drift (KED), Optimal Interpolation Method (OIM), Thiessen Polygons (TB) | Root Mean Squared Error (RMSE) | Optimal Interpolation Method (OIM) | [45] |
| Inverse Distance Weighting (IDW), Radial Basis Function (RBF), Diffusion Interpolation with Barrier (DIB), Kernel Interpolation with Barrier (KIB), Ordinary Kriging (OK), Empirical Bayesian Kriging (EBK) | Leave-One-Out Cross-Validation (LOOCV), Mean Square Error (MSE), Mean Absolute Error (MAE), Mean Absolute Percentage Error (MAPE), Symmetric Mean Absolute Percentage Error (SMAPE) Nash–Sutcliffe Efficiency Coefficient (NSE) | Kernel Interpolation with Barrier (KIB) | [46] |

**Table 1.** *Cont.*

| Interpolation Method | Validation Method | Recommended Method | Ref. |
|---|---|---|---|
| Inverse Distance Weighting (IDW), Radial Basis Function (RBF), Local Polynomial Interpolation (LPI), Global Polynomial Interpolation (GPI), Simple Kriging (SK), Universal Kriging (UK), Ordinary Kriging (OK), Empirical Bayesian Kriging (EBK), Empirical Bayesian Kriging Regression Prediction (EBKRP) | Mean Error (ME), Root Mean Square Error (RMSE), Pearson R2 (R2), Mean Standardized Error (MSE), Root Mean Square Standardized Error (RMSSE), Average Standard Error (ASE) | Empirical Bayesian Kriging Regression Prediction (EBKRP) | [47] |
| Inverse Distance Weighting (IDW), Kriging, ANUDEM, Spline | Mean Absolute Error (MAE), Mean Relative Error (MRE), Root Mean Squared Error (RMSE), Spatial and Temporal Distributions. | Inverse Distance Weighting (IDW) | [48] |
| Inverse Distance Weighting, Natural Neighbor (NN), Regularized Spline (RS), Tension Spline (TS), Ordinary Kriging (OK), Universal Kriging (UK) | Root Mean Square Error (RMSE), Mean Absolute Error (MAE), Mean Bias Error (MBE), Coefficient of Determination (R2) | Ordinary Kriging (OK) | [49] |
| Inverse Distance Weighting (IDW), Ordinary Kriging (OK), Ordinary Cokriging (OCK), Linear Regression (LR), Simple Kriging with varying Local Means (SKLM), Kriging with an External Drift (KED) | Mean Error (ME), and Root Mean-Square Error (RMSE) | Ordinary Cokriging (OCK) | [50] |
| Circular Ordinary Kriging (COK), Spherical Ordinary Kriging (SOK), Exponential Ordinary Kriging (EOK), Gaussian Ordinary Kriging (GOK), Empirical Bayesian Kriging (EBK) | Mean Error (ME), Mean Standardized Error (MSDE), Root Mean Square Standardized Error (RMSSDE) Mean Standard Error (MSE), Root Mean Square Error (RMSE) | Exponential Ordinary Kriging (EOK), Empirical Bayesian Kriging (EBK) | [51] |

Saudi Arabia ranks among the arid countries confronting challenges in accurately predicting rainfall patterns [4]. Therefore, this study aims to rectify the lack of rainfall data in an arid region, the Asir area in Saudi Arabia, to provide a reliable and accurate spatial distribution of rainfall. The objectives of this study encompass:

1. Assessing various spatial interpolation techniques to ascertain the optimal approach for accurate rainfall prediction across diverse arid regions.
2. Analyzing the sufficiency of rainfall station distribution and pinpointing optimal sites for installing new rain gauges within the study area.
3. Providing an illustrative example elucidating the practical utilization of study outcomes in filling data gaps at any location and time within the study area for end-users.

In this study, we collect rain gauge data for the Asir Province; however, some gauges lack rainfall records for certain years. As such, we first apply various spatial interpolation techniques to fill in the gap of the missing data for these gauges. Subsequently, we review the distribution of rain gauge stations to determine their sufficiency and determine if more meteorological stations are necessary for specific spots. Ultimately, we suggest the optimal locations for installing additional meteorological stations to improve data collection. The remainder of the article is organized as follows: Section 2 outlines the study area and its characteristics. Section 3 presents the materials and methods. Section 4 reports the main results and discussion of the findings. Section 5 demonstrates a use case to help

end-users apply the findings in practical situations, and the article ends with a summary and conclusions in Section 6.

## 2. The Study Area

The Kingdom of Saudi Arabia (KSA) is the world's 13th largest country, with an area of about 2,150,000 square kilometers [52,53]. The KSA extends between coordinates 33.75:56.25 E and 16.5:32.5 N, WGS84, and covers roughly 80% of the Arabian Peninsula [54]. The KSA overlooks both the Arabian Gulf and the Red Sea, boasting 550 and 2250 km of shoreline, respectively. These water bodies serve as the kingdom's primary sources of water vapor [55]. The KSA has a complex topography that can be classified into four categories: (1) coastal plains, (2) central and northern plateaus, (3) the central Tuwayq mountains, and (4) the western Asir mountains. Topographical elevations ascend steeply from the Red Sea shoreline towards the southwestern Asir mountains, reaching the kingdom's highest value at 2990 m above mean sea level [56]. The location and topography of the KSA are shown in Figure 1.

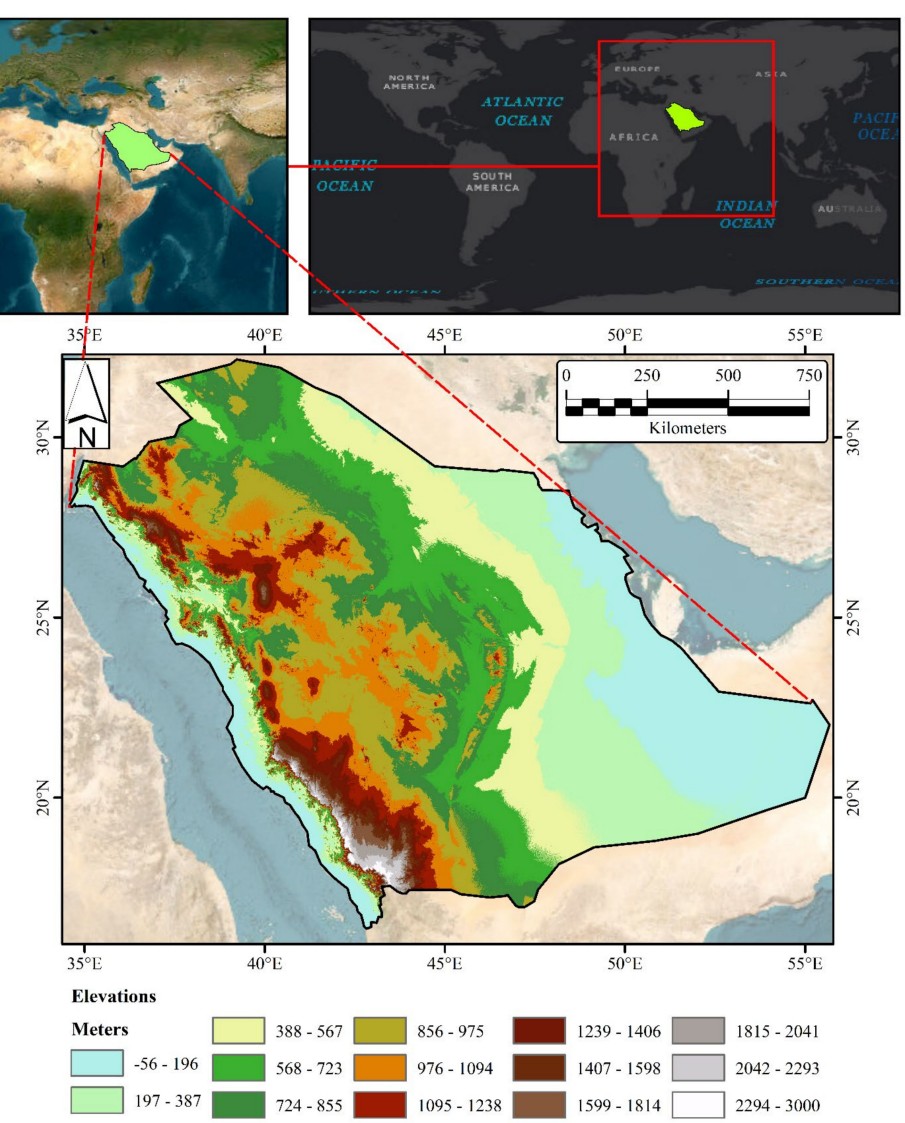

**Figure 1.** The KSA's location and topography.

The KSA is predominantly categorized into arid and semi-arid regions [57,58]. Given its prevailing arid climate and limited water resources, only five percent of cultivable land is utilized [59]. Notably, the majority of rainfall events occur between October and April [60]. Figure 2 shows the spatial distribution of the average annual rainfall over

the KSA, obtained from available rain gauge records [29]. The southwestern KSA region has the highest potential of harnessing rainfall as a water resource [60]. This region's mountainous terrain and complex topography generate orographic (convectional) rains, making it an ideal location for water resource exploitation [57,61,62]. Aligned with the World Bank's recommendations for addressing physical water resource scarcity within the MENA region [63], the area highlighted in Figure 2 was selected as the study area for this study.

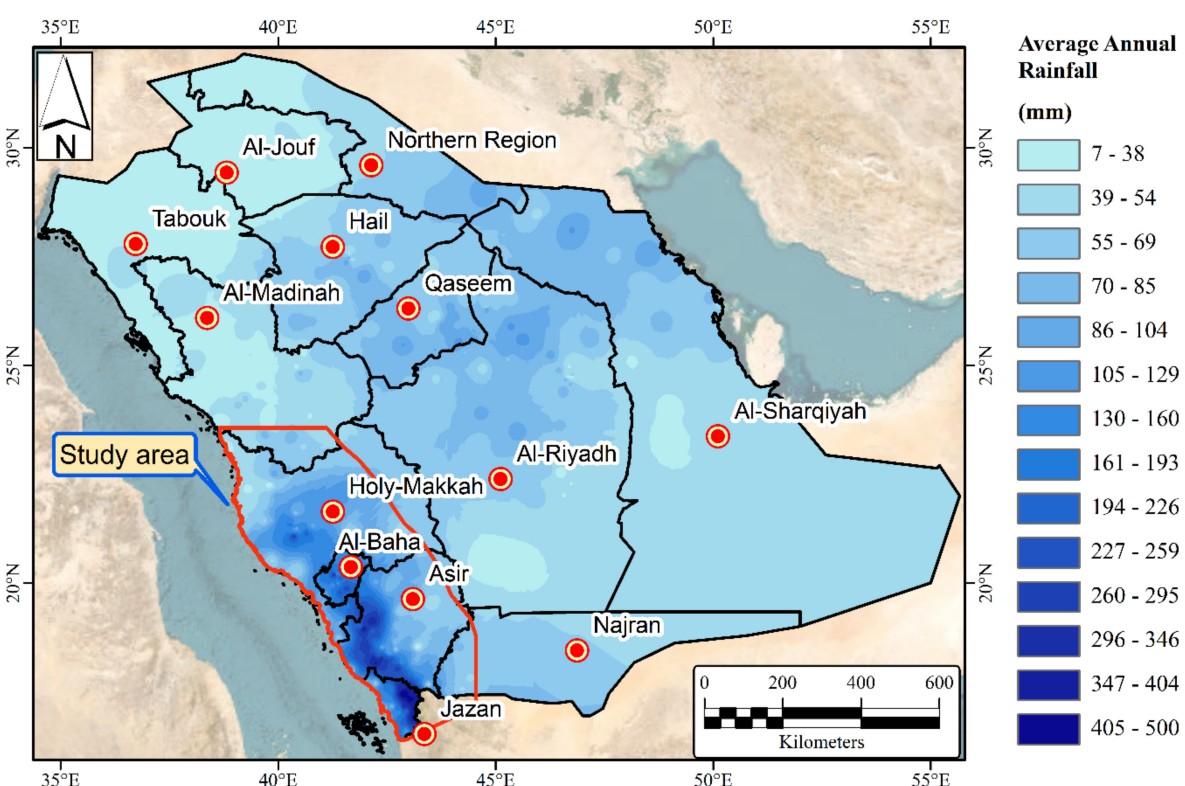

**Figure 2.** Average annual rainfall over the KSA, adopted from Helmi and Abdelhamed (2023) [29].

## 3. Materials and Methods

### 3.1. Rain Gauges

Assessing the adequacy of the rain gauge amount and distribution is the backbone of accurate quantification of rainfall volume. This imperative evaluation becomes particularly pronounced within mountainous terrains, owing to the pronounced spatial disparities in rainfall patterns. The available terrestrial rainfall data used in the current study were collected from official KSA authorities—the Ministry of Environment of the KSA (ME), the Ministry of Water and Agriculture of the KSA (MEWA), and the Presidency of Meteorology and Environment of the KSA (PME). Different types of rainfall gauges are available in the KSA: (1) meteorological (recording) gauges, (2) daily rainfall gauges, (3) recording rainfall gauges, and (4) totalizing rainfall gauges [64]. Within the geographical scope of this study, a total of 128 rain gauge data were acquired, as shown in Figure 3A. Notably, the available records obtained from totalizing gauges exhibit non-uniform temporal intervals, which can restrict their effectiveness in providing accurate estimates for interpreting/analyzing daily or yearly records. Five totalizing gauges are located in the study area, as highlighted in Figure 3B. Based on this criterion, the remaining 123 gauges were selected for the current study, as shown in Figure 3C. The study area has significant variability in altitude among its rain gauges, from 0 to 2603 m above mean sea level, as illustrated in Figure 3D. The properties of the selected rain gauges are summarized in Table 2, including the number of records available between the first and last available dates, as well as any missing yearly records (data gaps) during that period. Inconsistency can be noted in the available rain gauge data due to the randomness in rainfall patterns and the spatial variability in the

missing data locations in the available time frame. Figure 4 shows the temporal variation in the 123 selected gauges. To perform a geostatistical analysis in the study area, a threshold of at least 50 rain gauge data points was chosen. This threshold was met, allowing for a 48-year analysis period from 1966 to 2013. However, there are gaps in the rainfall records that need to be predicted with an appropriate interpolation method.

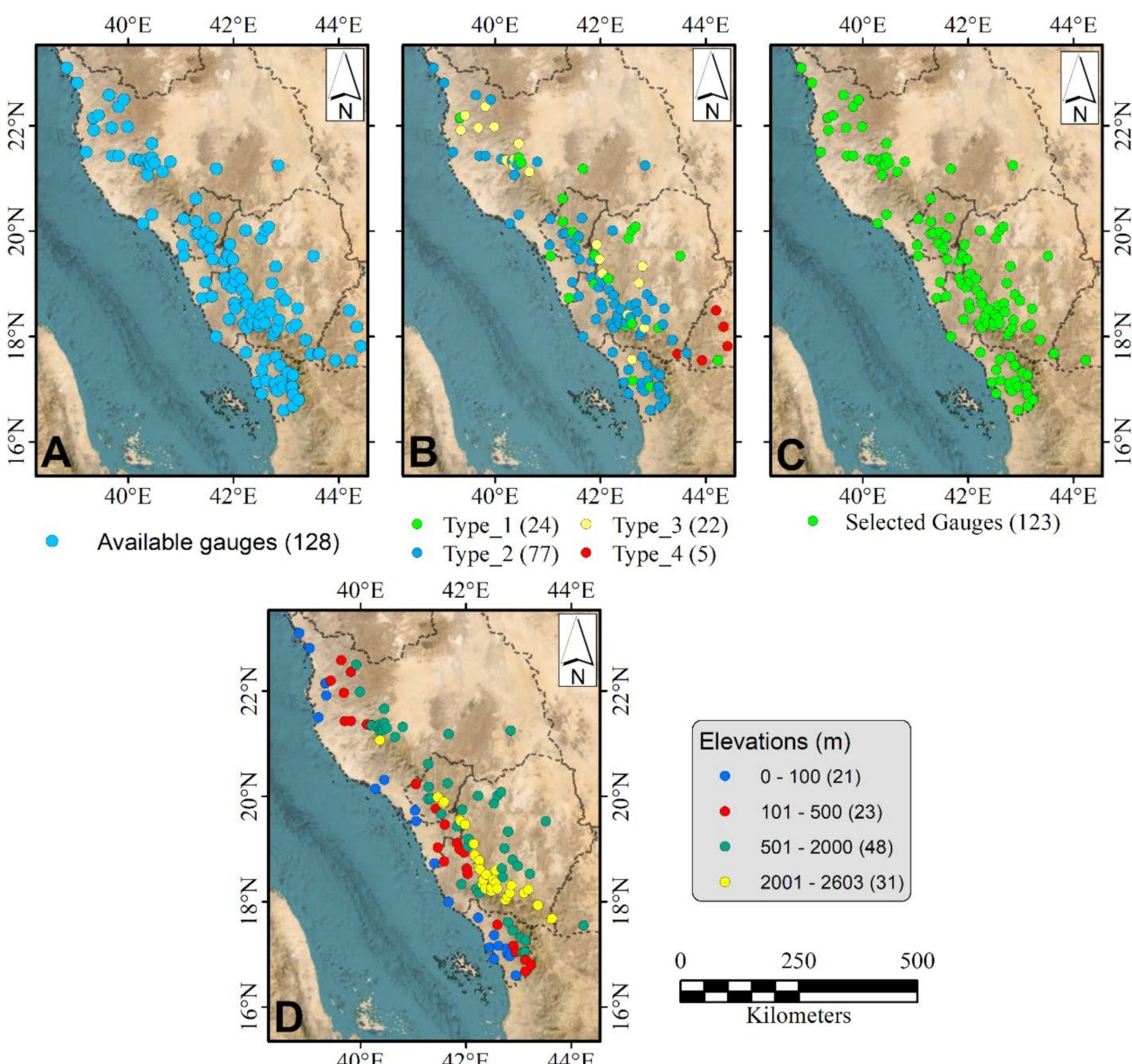

**Figure 3.** (**A**) The collected 128 rainfall gauges' locations, (**B**) types of available rainfall gauges, (**C**) the selected rainfall gauges for the current study, and (**D**) elevations of the selected rain gauges.

**Table 2.** Characteristics of the selected rain gauges.

| Gauge | Long (WGS 84) | Lat (WGS84) | Altitude (m) | Mean Max. Rainfall (mm) | Mean Total Rainfall (mm) | Gauge | Long (WGS 84) | Lat (WGS 84) | Altitude (m) | Mean Max. Rainfall (mm) | Mean Total Rainfall (mm) |
|---|---|---|---|---|---|---|---|---|---|---|---|
| A004 | 43.10 | 18.17 | 2280 | 31.9 | 97.9 | J141 | 39.92 | 22.50 | 661 | 49.7 | 96.4 |
| A005 | 42.48 | 18.20 | 2186 | 57.3 | 282.8 | J204 | 40.20 | 21.35 | 685 | 39.7 | 154.0 |
| A006 | 42.60 | 18.25 | 2121 | 40.3 | 238.5 | J205 | 40.22 | 21.35 | 750 | 46.3 | 171.0 |
| A007 | 42.15 | 19.10 | 2376 | 61.8 | 344.4 | J214 | 39.98 | 21.98 | 641 | 33.5 | 97.0 |
| A103 | 42.78 | 18.10 | 2224 | 39.5 | 237.9 | J219 | 39.43 | 22.20 | 192 | 22.2 | 38.4 |
| A104 | 43.37 | 17.93 | 2281 | 32.1 | 127.2 | J220 | 39.82 | 22.37 | 473 | 23.4 | 62.2 |
| A105 | 43.18 | 18.23 | 2206 | 23.9 | 70.1 | J221 | 39.35 | 21.92 | 88 | 27.5 | 45.0 |
| A106 | 42.48 | 18.27 | 2356 | 38.5 | 253.2 | J239 | 39.68 | 21.97 | 269 | 23.9 | 64.1 |
| A107 | 42.57 | 18.60 | 2016 | 31.9 | 109.8 | N001 | 44.23 | 17.55 | 1278 | 19.8 | 44.5 |
| A108 | 42.38 | 18.52 | 2516 | 26.4 | 85.0 | N103 | 43.63 | 17.68 | 2191 | 34.2 | 145.9 |
| A110 | 42.98 | 18.68 | 1802 | 33.9 | 98.3 | N203 | 43.62 | 17.67 | 2036 | 26.9 | 94.3 |
| A112 | 42.57 | 18.37 | 2096 | 36.2 | 112.4 | SA001 | 42.95 | 17.05 | 169 | 36.4 | 203.0 |
| A113 | 42.68 | 18.63 | 1840 | 17.5 | 69.3 | SA002 | 42.62 | 17.17 | 32 | 28.2 | 80.2 |
| A117 | 42.27 | 18.62 | 2489 | 38.2 | 181.7 | SA003 | 41.88 | 19.00 | 355 | 37.5 | 202.7 |
| A118 | 42.37 | 18.25 | 2197 | 56.7 | 333.7 | SA004 | 41.40 | 18.73 | 41 | 27.4 | 61.2 |
| A120 | 42.17 | 18.88 | 2308 | 56.0 | 213.9 | SA005 | 41.88 | 19.56 | 2263 | 27.2 | 172.5 |
| A121 | 42.75 | 18.03 | 2269 | 44.5 | 292.7 | SA101 | 42.83 | 16.97 | 68 | 41.6 | 173.0 |
| A123 | 42.87 | 18.32 | 2039 | 34.3 | 128.9 | SA102 | 42.23 | 17.70 | 51 | 30.1 | 65.2 |
| A124 | 42.33 | 18.42 | 2577 | 41.0 | 185.0 | SA104 | 43.08 | 17.04 | 574 | 50.3 | 403.9 |
| A126 | 43.22 | 18.53 | 1850 | 12.9 | 26.5 | SA105 | 41.97 | 18.93 | 458 | 48.6 | 364.1 |
| A127 | 42.25 | 18.78 | 2531 | 48.3 | 247.3 | SA106 | 42.53 | 17.37 | 73 | 37.2 | 137.5 |
| A128 | 42.70 | 18.47 | 1927 | 16.5 | 60.2 | SA107 | 42.78 | 17.12 | 78 | 44.6 | 170.0 |
| A130 | 42.32 | 18.33 | 2470 | 36.1 | 216.5 | SA108 | 41.92 | 18.33 | 533 | 41.8 | 255.8 |
| A201 | 42.52 | 18.42 | 2083 | 25.9 | 103.0 | SA110 | 43.13 | 17.27 | 1210 | 47.1 | 401.0 |

**Table 2.** *Cont.*

| Gauge | Long (WGS 84) | Lat (WGS84) | Altitude (m) | Mean Max. Rainfall (mm) | Mean Total Rainfall (mm) | Gauge | Long (WGS 84) | Lat (WGS 84) | Altitude (m) | Mean Max. Rainfall (mm) | Mean Total Rainfall (mm) |
|---|---|---|---|---|---|---|---|---|---|---|---|
| A206 | 42.25 | 18.68 | 2603 | 39.1 | 206.6 | SA111 | 43.12 | 17.05 | 574 | 54.5 | 444.1 |
| A213 | 42.83 | 18.17 | 2114 | 30.4 | 132.0 | SA113 | 42.03 | 18.53 | 455 | 41.3 | 298.1 |
| B001 | 41.29 | 20.18 | 1932 | 58.9 | 294.0 | SA115 | 41.67 | 18.00 | 0 | 28.1 | 54.6 |
| B004 | 42.60 | 20.02 | 1155 | 21.3 | 71.8 | SA116 | 42.20 | 18.25 | 1421 | 49.0 | 321.7 |
| B005 | 42.53 | 19.87 | 1202 | 18.6 | 73.6 | SA120 | 41.83 | 19.43 | 611 | 46.2 | 243.5 |
| B006 | 43.52 | 19.53 | 1090 | 21.8 | 48.9 | SA122 | 41.83 | 19.12 | 377 | 31.1 | 208.6 |
| B007 | 41.57 | 19.87 | 2047 | 52.6 | 269.3 | SA125 | 42.45 | 17.13 | 7 | 25.8 | 72.5 |
| B008 | 42.67 | 20.08 | 1139 | 24.3 | 80.1 | SA126 | 42.88 | 17.45 | 559 | 45.9 | 502.3 |
| B009 | 41.90 | 19.53 | 2279 | 46.0 | 272.6 | SA129 | 42.90 | 17.17 | 163 | 49.9 | 313.7 |
| B101 | 41.58 | 19.90 | 2040 | 50.8 | 224.9 | SA132 | 42.78 | 17.02 | 61 | 29.5 | 127.0 |
| B103 | 41.65 | 20.25 | 1571 | 20.0 | 51.5 | SA135 | 43.23 | 16.80 | 287 | 43.7 | 330.6 |
| B110 | 42.88 | 18.80 | 1742 | 22.4 | 73.2 | SA136 | 43.13 | 16.68 | 159 | 45.6 | 251.7 |
| B111 | 42.85 | 21.25 | 922 | 18.4 | 48.8 | SA137 | 42.95 | 16.60 | 61 | 29.5 | 116.3 |
| B114 | 42.23 | 20.02 | 1286 | 25.5 | 98.2 | SA138 | 42.02 | 18.63 | 393 | 39.7 | 267.1 |
| B208 | 42.73 | 19.02 | 1717 | 20.3 | 69.6 | SA139 | 42.03 | 19.05 | 900 | 26.2 | 293.3 |
| B216 | 41.98 | 19.47 | 2239 | 42.9 | 234.8 | SA140 | 43.03 | 17.32 | 688 | 49.7 | 448.6 |
| B217 | 41.93 | 19.75 | 1756 | 43.0 | 170.4 | SA142 | 41.58 | 18.77 | 104 | 32.3 | 101.9 |
| B219 | 42.80 | 19.33 | 1475 | 18.4 | 53.1 | SA143 | 43.13 | 16.90 | 259 | 46.3 | 420.7 |
| B220 | 42.04 | 19.20 | 1571 | 36.0 | 142.6 | SA144 | 42.25 | 18.17 | 1013 | 46.9 | 366.6 |
| J001 | 41.05 | 19.53 | 53 | 36.8 | 77.3 | SA145 | 42.80 | 17.62 | 699 | 46.0 | 306.8 |
| J002 | 39.34 | 22.16 | 72 | 18.3 | 33.3 | SA147 | 41.47 | 19.03 | 115 | 30.9 | 66.3 |
| J102 | 39.70 | 21.43 | 355 | 27.8 | 50.3 | SA148 | 42.53 | 16.92 | 0 | 35.5 | 79.0 |
| J106 | 39.33 | 22.15 | 66 | 18.8 | 35.8 | SA204 | 42.60 | 17.57 | 188 | 43.3 | 160.2 |
| J107 | 40.45 | 20.32 | 95 | 34.1 | 63.6 | TA002 | 40.50 | 21.30 | 1590 | 33.0 | 148.2 |
| J108 | 40.28 | 20.15 | 7 | 32.8 | 59.8 | TA004 | 40.45 | 21.40 | 1553 | 28.6 | 119.4 |
| J111 | 38.83 | 23.10 | 12 | 17.4 | 33.7 | TA005 | 41.67 | 21.18 | 1148 | 14.0 | 49.4 |

**Table 2.** *Cont.*

| Gauge | Long (WGS 84) | Lat (WGS84) | Altitude (m) | Mean Max. Rainfall (mm) | Mean Total Rainfall (mm) | Gauge | Long (WGS 84) | Lat (WGS 84) | Altitude (m) | Mean Max. Rainfall (mm) | Mean Total Rainfall (mm) |
|---|---|---|---|---|---|---|---|---|---|---|---|
| J113 | 40.12 | 21.37 | 455 | 47.8 | 146.0 | TA006 | 41.28 | 20.62 | 1389 | 52.6 | 117.9 |
| J114 | 39.82 | 21.43 | 298 | 37.9 | 78.6 | TA007 | 41.47 | 19.98 | 2256 | 48.6 | 194.0 |
| J116 | 39.63 | 22.58 | 394 | 28.2 | 63.3 | TA104 | 40.80 | 21.32 | 1394 | 31.4 | 97.3 |
| J121 | 41.05 | 20.23 | 338 | 40.9 | 162.1 | TA106 | 40.32 | 21.33 | 1888 | 44.8 | 185.5 |
| J124 | 41.28 | 19.95 | 586 | 30.0 | 114.4 | TA109 | 40.37 | 21.07 | 2145 | 45.4 | 269.6 |
| J126 | 41.43 | 19.77 | 353 | 37.3 | 236.8 | TA125 | 40.42 | 21.26 | 1713 | 29.7 | 51.7 |
| J127 | 41.53 | 19.67 | 657 | 52.7 | 184.0 | TA206 | 40.40 | 21.28 | 1675 | 35.7 | 165.8 |
| J131 | 41.60 | 19.47 | 474 | 41.9 | 182.6 | TA233 | 40.65 | 21.13 | 1691 | 48.5 | 205.6 |
| J134 | 39.20 | 21.50 | 15 | 28.7 | 51.4 | TA250 | 40.45 | 21.67 | 1241 | 22.7 | 91.2 |
| J137 | 41.33 | 19.97 | 632 | 36.4 | 222.3 | TA251 | 40.37 | 21.37 | 1822 | 30.1 | 120.1 |
| J139 | 41.03 | 19.73 | 93 | 38.6 | 63.9 | TA255 | 40.36 | 21.24 | 1730 | 34.1 | 118.5 |
| J140 | 39.03 | 22.82 | 10 | 21.1 | 39.4 | | | | | | |

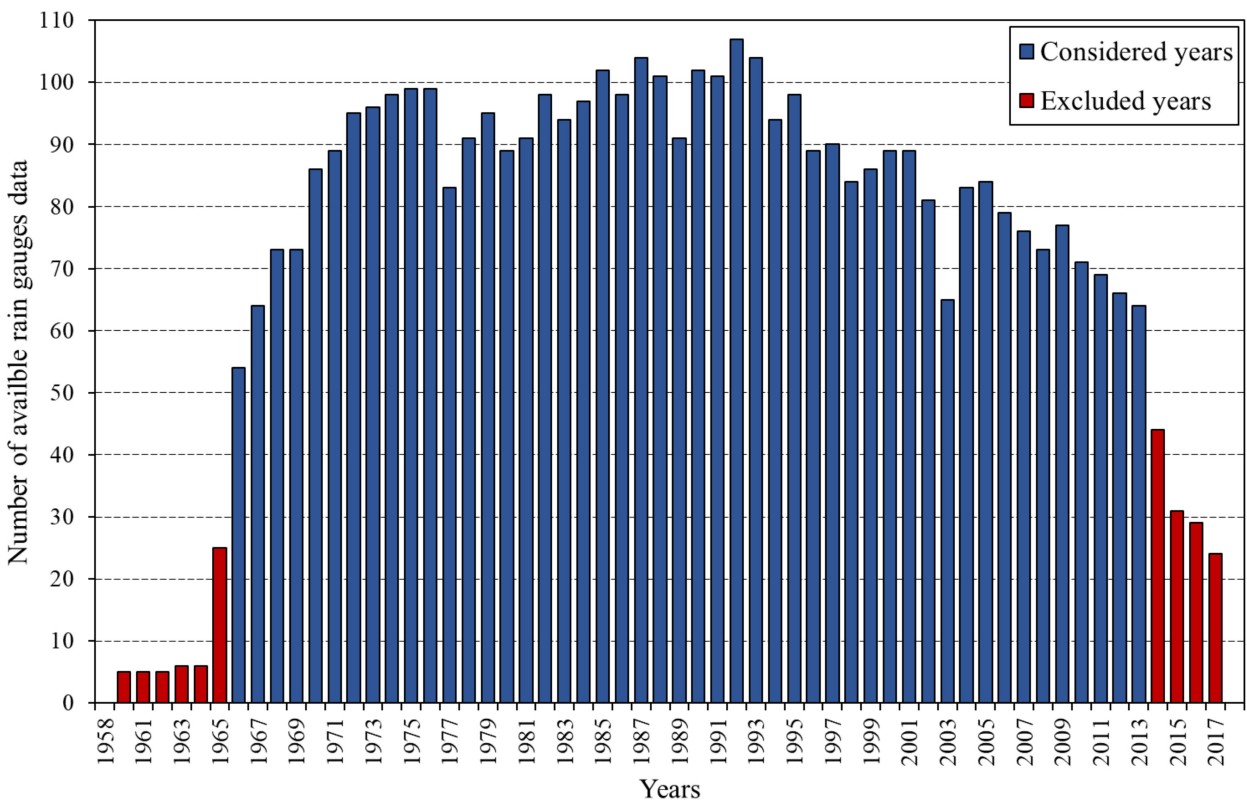

**Figure 4.** Temporal variation in available rain gauge data in the study area.

*3.2. Interpolation Techniques*

The characteristics of available datasets influence the spatial interpolation outcomes, so selecting an accurate interpolation method is a crucial part of generating accurate distribution rainfall maps. Until recently, most of the studies in this field provided a limited comparison of interpolation techniques, relying on statistical errors to select the best method. The choice of interpolation method depends on the nature of the data, the size of the study area, and the desired level of accuracy. Each method has its pros and cons, and it is important to choose the most appropriate one for a specific application [65].

The selected spatial interpolation techniques were systematically analyzed to determine their accuracy in estimating unknown rainfall values and predicting the spatial rainfall distribution pattern. In the current study, the authors selected to evaluate two different interpolation categories, i.e., deterministic and geostatistical. Nearly all commonly known interpolation techniques and various models within these techniques were analyzed to determine the most optimal method for interpolating rainfall data in the study area.

Four interpolation techniques were evaluated under the deterministic technique umbrella: Inverse Distance Weighting (IDW), Global Polynomial Interpolation (GPI), Radial Basis Functions (RBF), and Local Polynomial Interpolation (LPI). Additionally, three interpolation techniques under the geostatistical technique umbrella were also evaluated, which were the Kriging, Cokriging, and Empirical Bayesian Kriging (EBK), with seven different semivariogram models such as Circular, Spherical, Exponential, Gaussian, K-Bessel, J-Bessel, and Stable models. The interpolation techniques' mathematical equations are given in Appendix A.

Generally, 51 interpolation techniques (43 geostatistical and 8 deterministic) were selected to be assessed in the current study to determine the most suitable interpolation technique for the data available, as shown in Table 3.

**Table 3.** Codes of the 51 interpolation techniques used in the current study.

| Geostatistical Interpolation Techniques | |
| --- | --- |
| Ordinary Kriging—Circular Variogram (KOC) | Ordinary Cokriging—Circular Variogram (CKOC) |
| Ordinary Kriging—Spherical Variogram (KOS) | Ordinary Cokriging—Spherical Variogram (CKOS) |
| Ordinary Kriging—Exponential Variogram (KOE) | Ordinary Cokriging—Exponential Variogram (CKOE) |
| Ordinary Kriging—Gaussian Variogram (KOG) | Ordinary Cokriging—Gaussian Variogram (CKOG) |
| Ordinary Kriging—K-Bessel Variogram (KOK) | Ordinary Cokriging—K-Bessel Variogram (CKOK) |
| Ordinary Kriging—J-Bessel Variogram (KOJ) | Ordinary Cokriging—J-Bessel Variogram (CKOJ) |
| Ordinary Kriging—Stable Variogram (KOT) | Ordinary Cokriging—Stable Variogram (CKOT) |
| Simple Kriging—Circular Variogram (KSC) | Simple Cokriging—Circular Variogram (CKSC) |
| Simple Kriging—Spherical Variogram (KSS) | Simple Cokriging—Spherical Variogram (CKSS) |
| Simple Kriging—Exponential Variogram (KSE) | Simple Cokriging—Exponential Variogram (CKSE) |
| Simple Kriging—Gaussian Variogram (KSG) | Simple Cokriging—Gaussian Variogram (CKSG) |
| Simple Kriging—K-Bessel Variogram (KSK) | Simple Cokriging—K-Bessel Variogram (CKSK) |
| Simple Kriging—J-Bessel Variogram (KSJ) | Simple Cokriging—J-Bessel Variogram (CKSJ) |
| Simple Kriging—Stable Variogram (KST) | Simple Cokriging—Stable Variogram (CKST) |
| Universal Kriging—Circular Variogram (KUC) | Universal Cokriging—Circular Variogram (CKUC) |
| Universal Kriging—Spherical Variogram (KUS) | Universal Cokriging—Spherical Variogram (CKUS) |
| Universal Kriging—Exponential Variogram (KUE) | Universal Cokriging—Exponential Variogram (CKUE) |
| Universal Kriging—Gaussian Variogram (KUG) | Universal Cokriging—Gaussian Variogram (CKUG) |
| Universal Kriging—K-Bessel Variogram (KUK) | Universal Cokriging—K-Bessel Variogram (CKUK) |
| Universal Kriging—J-Bessel Variogram (KUJ) | Universal Cokriging—J-Bessel Variogram (CKUJ) |
| Universal Kriging—Stable Variogram (KUT) | Universal Cokriging—Stable Variogram (CKUT) |
| Empirical Bayesian Kriging (EBK) | |
| Deterministic interpolation techniques | |
| Global Polynomial Interpolation (GPI) | Inverse Distance Weighting—P = 2 (IDWP2) |
| Local Polynomial Interpolation (LPI) | Inverse Distance Weighting—P = 3 (IDWP) |
| Radial Basis Function (RBF) | Inverse Distance Weighting—P = 4 (IDWP4) |
| Inverse Distance Weighting (P = 1) (IDWP1) | Inverse Distance Weighting—P = 5 (IDWP) |

### 3.2.1. Deterministic Techniques

The Inverse Distance Weighting (IDW) method estimates the value of the target point based on the values of neighboring points, with more weight given to points closer to the target point. The IDW method uses the Cartesian coordinates of the target station and uses the inverse distance raised to a power P as a weighting factor for the adjacent points values, as shown in Equation (1) [31,33,34,36,42]. According to this method, the influence of a variable diminishes with the distance from its sampled location. Using a high power in IDW emphasizes the nearest points, and creates a more detailed surface with lower smoothness [66]. Five IDW interpolation powers were tested in the current study, namely IDWP1, IDWP2, IDWP3, IDWP4, and IDWP5, for $P$ = 1, 2, 3, 4, and 5, respectively.

$$z_p = \frac{\sum_{i=1}^{n}\left(\frac{z_i}{d_i{}^p}\right)}{\sum_{i=1}^{n}\left(\frac{1}{d_i{}^p}\right)} \tag{1}$$

where $z_i$ is the known value point, $di$ is the distance to a known point, $z_i$ is the unknown point, and $p$ is the exponent power (1, 2, 3, 4, or 5) [67].

Global Polynomial Interpolation (GPI) is a mathematical technique that can calculate a polynomial function to pass through a set of given points. GPI is a highly accurate tool in numerical analysis and can be used to estimate any continuous function. The key advantage of GPI over other interpolation techniques is that it provides a single polynomial function to approximate the entire curve instead of several piecewise functions. To perform GPI, a set of points that lie on the curve to be estimated must be chosen, and a polynomial function is then constructed to pass through these points. The degree of the polynomial is determined by the number of points selected. Higher-degree polynomials can provide more accurate approximations, but they may also be more prone to rounding errors and other factors. GPI is a fast and global method of predicting rainfall based on the polynomial function [66,68]. The second-degree polynomial is selected for the current study.

The Radial Basis Function (RBF), also known as the Spline method, uses artificial neural networks (ANN) to accurately interpolate data. The RBF is a popular tool for interpolating multi-dimensional scattered data and can be easily generalized to various space dimensions, making it useful in natural resource management. In addition to acting as a class of interpolation techniques for georeferenced data, the RBF is a deterministic interpolator that utilizes the level of smoothing to determine the appropriate interpolation technique [68,69].

Local Polynomial Interpolation (LPI) is a method for approximating a function by fitting a polynomial to small subsets of the data. This technique involves constructing a locally best-fitting polynomial to a set of points, rather than fitting a single polynomial to all the points globally. LPI is particularly useful when dealing with large datasets or highly variable functions. By considering the local behavior of the data, it allows for a more accurate representation of the function. In addition, LPI can estimate derivatives and integrals of the function at specific points [68].

### 3.2.2. Geostatistical Techniques

Geostatistics is a statistical technique that deals with spatially distributed variables. Its original purpose was to predict the likelihood of finding gold ore in the Witwatersrand mine in South Africa [70]. This technique employs a semivariogram, which describes the spatial correlation between data points, to estimate values at locations where no samples exist. The semivariogram, which plots the variance of the differences between pairs of points against their spatial separation, is vital to geostatistical interpolation. It is used to model the spatial dependence of the estimated variables and to interpolate values at unsampled locations. The semivariogram provides information about the spatial correlation between data points and is used to estimate the weights that are applied to the measured values of the variable being estimated at unsampled locations [68].

Kriging is a widely used technique in the fields of geology, environmental science, and engineering to estimate the value of a variable at an unobserved location based on nearby observed locations. The technique relies on spatial autocorrelation, which refers to the tendency of locations to exhibit similar values to the given variable. Kriging utilizes this principle to estimate the variable's value at an unobserved location by analyzing the spatial correlation between nearby observed locations. This involves fitting a mathematical model to the observed data, which describes the spatial correlation between data points. The resulting model is then used to estimate the variable's value at the unobserved location by taking a weighted average of the values of the variable at nearby observed locations, with the weights assigned based on the spatial correlation between the locations [21,31,33,42]. Kriging offers several benefits over other interpolation techniques, including more accurate estimates and a measure of uncertainty for each estimate. This can be particularly useful in decision-making processes, where a clear understanding of the accuracy and reliability of the estimates is critical. The semivariogram is a mathematical function used to describe the spatial autocorrelation of a variable and is used to measure the degree to which the values of the variable at two locations are similar as a function of the distance between these locations.

Cokriging is a variation of Kriging that enables the inclusion of additional spatially correlated variables in the estimation process. This method enhances the precision of rainfall estimations by considering the connection between rainfall and other factors such as elevation, temperature, and land use. It is particularly beneficial when the rainfall data are limited or unreliable. The Cokriging method assesses the cross-covariance function between two variables, describing their spatial correlation. It then uses this function to determine the weights to be assigned to the measured values of the related variable to predict the rainfall values at unsampled locations. Cokriging is a valuable tool for spatially analyzing rainfall, improving our understanding and prediction of rainfall patterns [71,72].

Empirical Bayesian Kriging (EBK) is a geostatistical interpolation method used to estimate rainfall distribution in spatial domains. It tackles some challenges present in spatially valid Kriging models, such as the manual adjustment of parameters required by other Kriging techniques. Instead, EBK automates the parameter calculation process by combining submissions and simulations [73–75]. EBK stands out from other Kriging techniques due to its ability to account for errors arising from semivariogram estimation. It employs a hierarchical Bayesian model that incorporates prior semivariogram knowledge, enhancing the accuracy of interpolated values by reducing the estimation errors. This flexible approach yields accurate estimates, especially with limited data. The EBK process includes estimating a semivariogram model from available data, simulating new values at input locations based on this model, and iteratively estimating new semivariogram models from simulated data. This produces a spectrum of semivariograms, enabling prediction and standard error computation for unsampled locations. In contrast to classical Kriging, EBK optimizes parameters differently. It automatically refines parameters using diverse semivariogram models, resulting in enhanced adaptability and estimate accuracy [73–75].

## 4. Results and Discussion

Automation of geostatistical analysis in Arc-GIS via template parameters can be incorporated into deterministic and geostatistical approaches that do not require semivariograms. Seven types of semivariograms (i.e., Circular, Spherical, Exponential, Gaussian, K-Bessel, J-Bessel, and Stable) were selected to be evaluated within Kriging and Cokriging techniques and their sub-types (i.e., ordinary, simple, and universal). A semivariogram relates the distance between rain gauge locations and the variance between rain gauge records. Consequently, the semivariogram varies from year to year based on available records. These variances and inability to automate the process necessitate the manual generation of 4032 geostatistical parameters for Kriging and Cokriging through the 48-year study period (2 (Kriging and Cokriging) × 3 (Kriging types) × 7 (semivariogram types) × 48 (number of years)). For other interpolation techniques, parameter values (e.g., the power of the IDW) were defined once and repeated for each year of the study period.

Several quantitative statistical metrics are found in the literature, offering different lenses for evaluating the accuracy of interpolation techniques. The Mean Absolute Error (MAE: Equation (2)) measures the average error between predicted ($P_i$) and measured rainfall values ($M_i$). The Root Mean Square Error (RMSE: Equation (3)) calculates the standard deviation of the errors. The Centered Root Mean Square Error (CRMSE: Equation (4)) disregards the mean values in the error evaluation. Taylor's graphical plot (Taylor, 2001) provides valuable information about the standard deviation, the correlation coefficient (CC: Equation (5)), and CRMSE for ($M_i$) and ($P_i$).

In this study, the cross-validation CRMSE serves as the criterion for comparing the prediction accuracy across different geostatistical approaches, guiding the selection of the appropriate technique with the least CRSME. This cross-validation procedure was sequentially conducted across all the points used to generate a geostatistical layer. After omitting each measured value from the dataset, the error at each point was determined by the difference between the predicted and the measured values at the respective location. Across the study period, distinct geostatistical approaches were assigned to each year, addressing total yearly and maximum daily rainfall data. This variation in the appropriate

geostatistical approaches is due to the differences in the available rain gauges used to create the dataset from one year to another. Figure 5 shows the Taylor diagram for total yearly rainfall in the study area for the year 1998. Among the approaches, ordinary Cokriging with an exponential variogram (CKOE) emerged with the least centered Root Mean Square Error and thus was selected to generate the total yearly surface for the study area. Similarly, for the year 1999, the spatial distribution of the total annual and maximum daily rainfall depths over the study area based on the selected (CKSG and CKOC) interpolation techniques is given in Figures 6 and 7, respectively. Table 4 shows the yearly selected interpolation techniques for the maximum daily and total annual rainfall over the study area.

$$\text{MAE} = \frac{\sum_{i=1}^{N} |(M_i - P_i)|}{N} \tag{2}$$

$$\text{RMSE} = \sqrt{\frac{1}{N} \sum_{i=1}^{N} (M_i - P_i)^2} \tag{3}$$

$$\text{CRMSE} = \sqrt{\frac{1}{N} \sum_{i=1}^{N} \left[ (M_i - \overline{M}) - (P_i - \overline{P}) \right]^2} = \sqrt{\sigma_M^2 + \sigma_P^2 - 2 \cdot \sigma_P \cdot \text{CC}} \tag{4}$$

$$\text{CC} = \frac{\sum_{i=1}^{n} (M_i - \overline{M}) \cdot (P_i - \overline{P})}{\sqrt{\sum_{i=1}^{n} (M_i - \overline{M})^2} \cdot \sqrt{\sum_{i=1}^{n} (P_i - \overline{P})^2}} \tag{5}$$

where $\mu_S$ and $\sigma_S$ are the mean and standard deviation of satellite rainfall data.

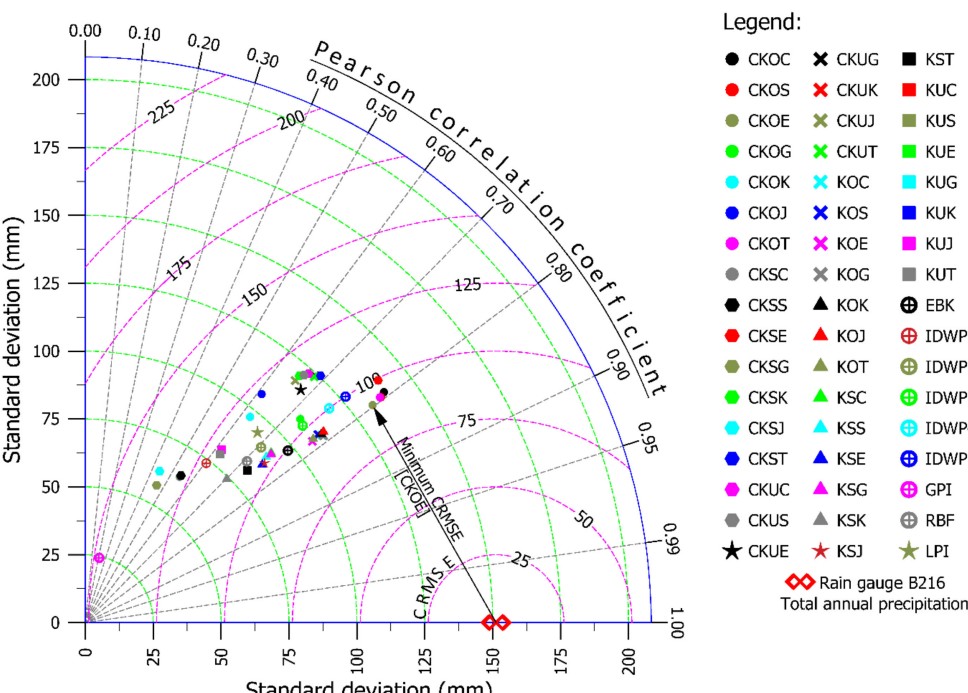

**Figure 5.** Taylor diagram depicting total rainfall interpolation models for the year 1998.

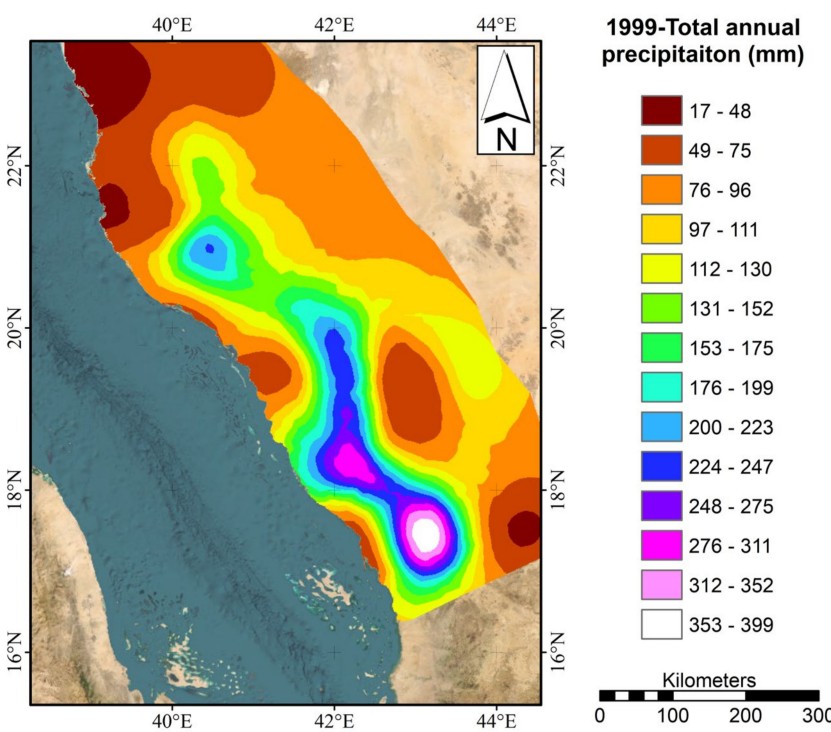

**Figure 6.** Spatial distribution of total annual rainfall depth for the year 1999 (generated by CKSG geostatistical interpolation).

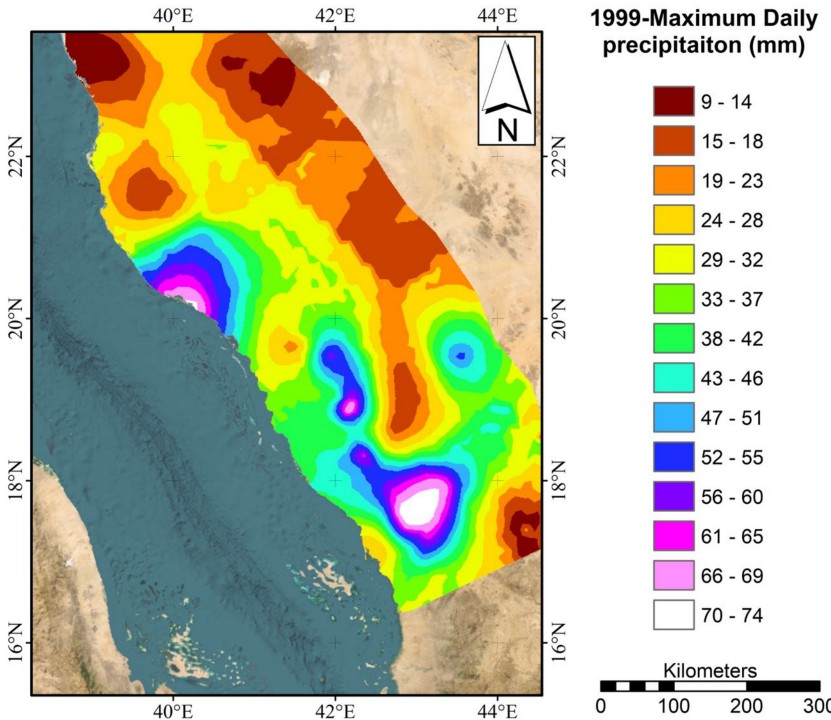

**Figure 7.** Spatial distribution of maximum daily rainfall for the year 1999 (generated by CKOC geostatistical interpolation).

**Table 4.** Yearly selected interpolation techniques for maximum daily and total annual rainfall over the study area.

| Year | Total Yearly | Maximum Daily | Year | Total Yearly | Maximum Daily |
|------|--------------|---------------|------|--------------|---------------|
| 1966 | KOJ | CKOK | 1990 | CKUK | CKOJ |
| 1967 | KUJ | KUJ | 1991 | CKSJ | KOJ |
| 1968 | KSJ | CKUC | 1992 | CKOJ | KSG |
| 1969 | LPI | CKOJ | 1993 | CKUJ | CKSK |
| 1970 | KSJ | KUJ | 1994 | CKOS | KOJ |
| 1971 | CKOC | CKOC | 1995 | CKOC | CKUC |
| 1972 | CKOC | KOJ | 1996 | CKSE | CKOG |
| 1973 | CKOJ | KOJ | 1997 | CKUT | CKOT |
| 1974 | KOJ | CKOC | 1998 | CKUK | CKOK |
| 1975 | CKOJ | CKUJ | 1999 | CKSG | CKOC |
| 1976 | CKOJ | LPI | 2000 | KSG | KOE |
| 1977 | CKOG | CKUC | 2001 | KOT | KSJ |
| 1978 | CKOJ | KUJ | 2002 | CKSC | CKOG |
| 1979 | CKOT | CKOS | 2003 | KUJ | CKSJ |
| 1980 | KOJ | CKOJ | 2004 | KOJ | CKOJ |
| 1981 | CKSG | CKSJ | 2005 | CKSC | GPI |
| 1982 | CKSG | CKUJ | 2006 | KOG | CKOK |
| 1983 | CKUC | KOJ | 2007 | KOG | IDWP1 |
| 1984 | CKOC | CKSK | 2008 | CKSJ | CKSG |
| 1985 | CKOE | CKOK | 2009 | CKSE | KUJ |
| 1986 | CKOC | KUJ | 2010 | KSG | CKOC |
| 1987 | KOJ | CKSK | 2011 | KSJ | KST |
| 1988 | KUK | CKOC | 2012 | CKOS | CKUJ |
| 1989 | CKUE | LPI | 2013 | CKOJ | CKOC |

The yearly spatial distribution for maximum and total precipitation over the study area with gaps filled from 1966 to 2013 is given in Figure S1 through Figure S12 in the Supplementary Materials. These rasters can be utilized to directly select any missing data at a specific gauge within the study area boundaries and during the study period (as shown in the use case section). The average values of the total and maximum yearly precipitation were calculated based on the filled data from 1966 to 2013, as given in Figure 8. These average values were utilized to select the optimal interpolation technique in light of the limited availability of rainfall data beyond 2013 due to the limited rain gauge records in this period. In such cases, the Ordinary Kriging with J-Bessel semivariogram (KOJ) and Simple Kriging with K-Bessel (KSK) semivariogram techniques are recommended to fill data gaps for maximum and total yearly precipitation, respectively, for the years after 2013.

The World Meteorological Organization (WMO) recommends certain densities of rain gauge stations to be followed for different types of catchments. For mountainous regions with irregular rainfall, a density of 250 km$^2$ per station is recommended for daily-recording gauges [23]. The density of the rain gauges in the study area is far beyond these recommended limits. Additionally, several publications have discussed different techniques for assessing and optimizing the locations of rain gauge stations and the spatial distribution of the network. These techniques include statistical covariance [76], the Kriging

interpolation method, and Kriging and entropy techniques for rainfall network design [77]. In this study, the cross-validation error (CVE) is used as a quality indicator to evaluate the density of a given rain gauge network. The spatial distribution of the CVE for the global proposed maximum daily and total yearly interpolation techniques (KOJ, KSK) is displayed in Figures S13A and S14A in the Supplementary Materials, with the sum of both errors shown in Figure 9A. The maximum combined CVE resulting from the sum of the total rainfall and the maximum rainfall is greater than 50%. The highest CVE is observed in the southeastern part of the study area, likely due to orographic effects. The average of the combined CVE is 42%, as highlighted in Table 5.

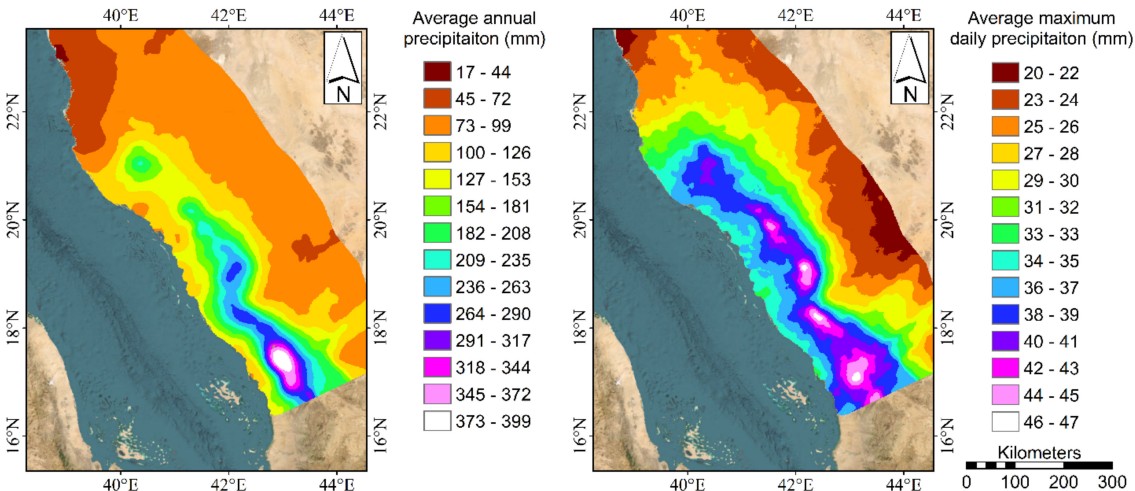

**Figure 8.** The spatial distribution of the maximum and total yearly precipitation (mm).

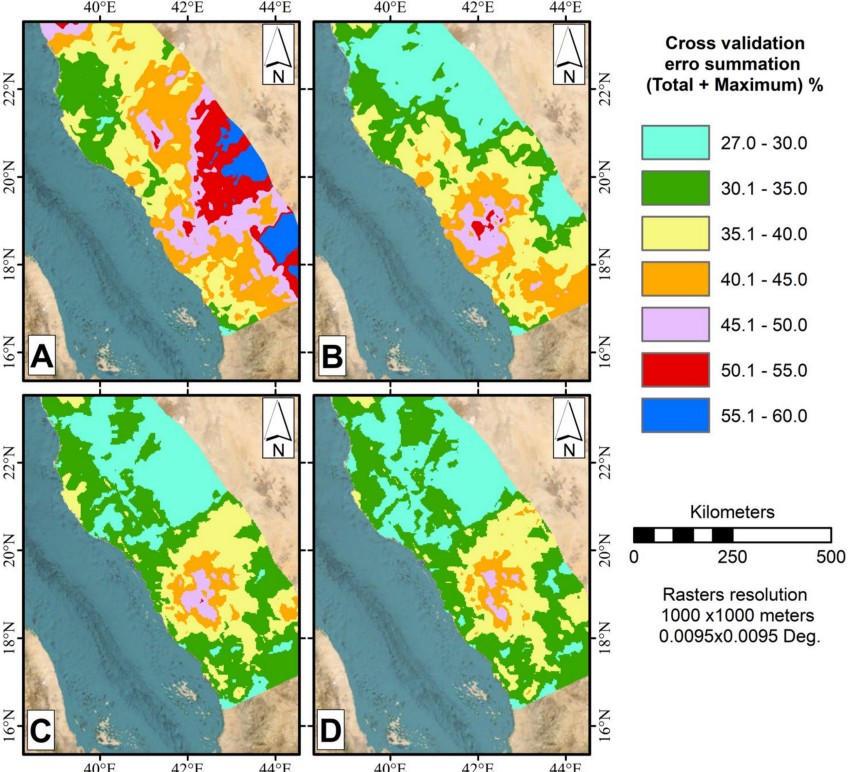

**Figure 9.** Cross-validation error summation for total yearly and maximum daily rainfall for (**A**) existing rainfall stations, (**B**) with additional proposed stations (I), (**C**) with additional proposed stations (I) and (II), and (**D**) with additional proposed stations (I), (II), and (III).

**Table 5.** Study area areal average cross-validation error (%).

| | Maximum Rainfall Error (%) | Total Rainfall Error (%) | Error Summation (%) |
|---|---|---|---|
| Existing stations | 14.60% | 27.08% | 41.68% |
| Existing stations + Proposed stations (I) | 12.71% | 22.34% | 35.05% |
| Existing stations + Proposed stations (I) and (II) | 11.65% | 20.95% | 32.61% |
| Existing stations + Proposed stations (I), (II), and (III) | 11.51% | 20.79% | 32.31% |

Three additional groups of rain gauges (I, II, and III) are proposed to reduce the CVE, resulting in a total of 135, 148, and 153 stations, respectively, as shown in Figure 10. The proposed locations of the additional rain gauges are identified based on three criteria: (1) areas with the highest cross-validation error, (2) areas with a low rain gauge density, and (3) the proposed locations should be selected to avoid screening other stations located at centroid of the triangle formed by the three closest gauges. Figure 11 demonstrates the proposed methodology for locating new stations.

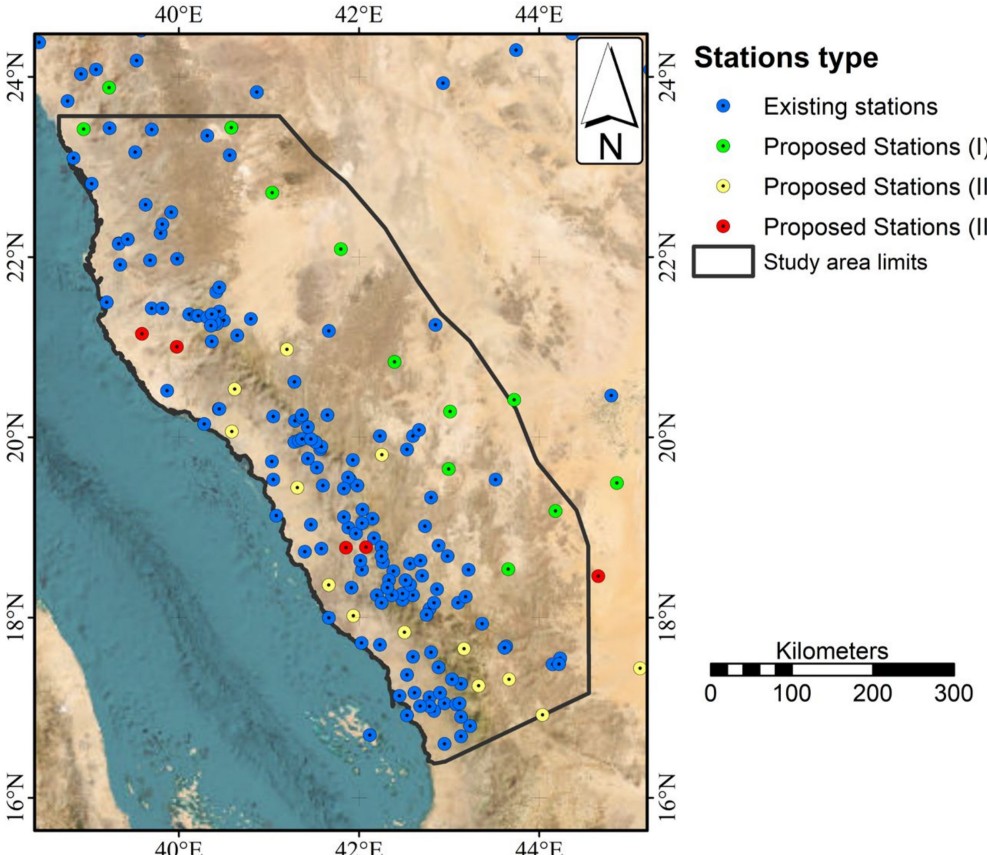

**Figure 10.** Proposed additional stations.

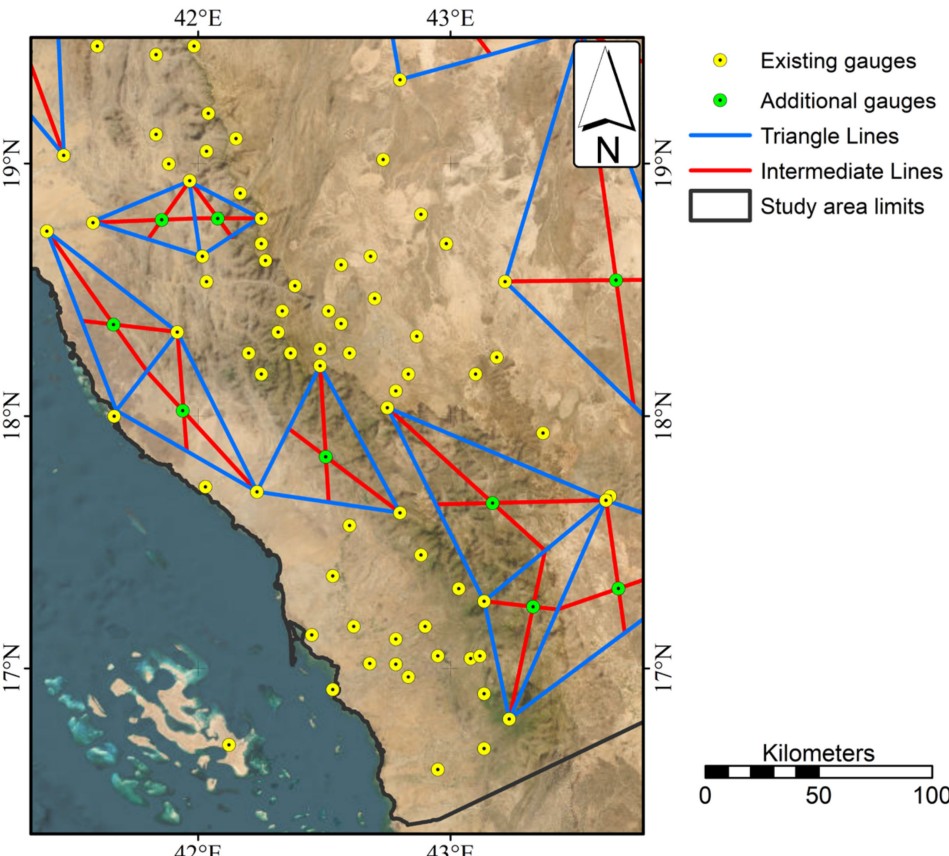

**Figure 11.** Proposed methodology to select the location of additional stations.

Table 5 shows that adding new candidate stations generally helped to decrease the average CVE. It is interesting to note that this reduction in CVE is not linear, as shown in Figure 12. Increasing the existing rain gauge stations by 10% (group I) resulted in a decrease in the average CVE by 16%. However, the additional addition of 10% of the number of rain gauge stations from group I to group II resulted in a decrease of less than 6% in the average CVE.

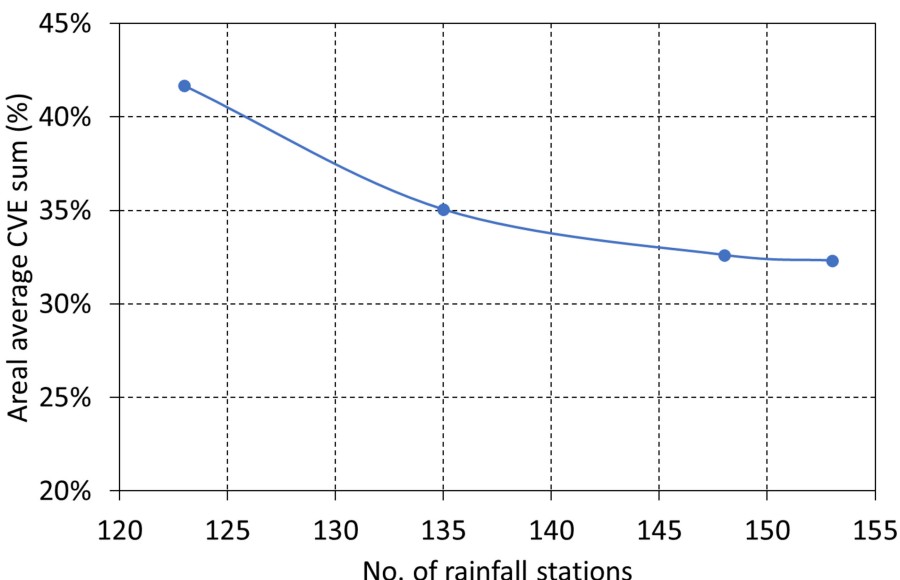

**Figure 12.** Effect of increasing the number of rain gauge stations on the average CVE.

Assessing and categorizing the aridity in different climatic regions have significant implications for agricultural productivity, water resource management, and environmental conservation. The United Nations Environment Programme (UNEP) introduced the Aridity Index (AI) as a quantitative measure of aridity, involving the evaluation of the ratio of annual rainfall (P) to potential evapotranspiration (PET) [78]. The aridity is classified as desert, hyper-arid, arid, semi-arid, dry subhumid, or subhumid based on the AI values shown in Table 6 [79]. Hyper-arid regions receive minimal rainfall. Arid regions have insufficient rainfall to sustain vegetation and semi-arid regions experience slightly higher but still inadequate rainfall, while dry, sub-humid regions receive enough rainfall to support continuous harvesting without requiring irrigation.

**Table 6.** Aridity level classification.

| Aridity Level | Aridity Index (AI) |
|---|---|
| Desert | AI < 0.03 |
| Hyper arid | 0.03 < AI < 0.05 |
| Arid | 0.05 < AI < 0.20 |
| Semi-arid | 0.20 < AI < 0.50 |
| Dry | 0.50 < AI < 0.65 |
| Sub-humid | 0.65 < AI < 0.75 |
| Humid | AI > 0.75 |
| Cold | PET $\leq$ 400 mm |

To determine the AI within the study area, it was necessary to determine the average annual rainfall and PET values. Figure 8 shows that the time-averaged annual rainfall varies from a minimum of 17 mm/year (at the northwestern boundary of the study area) to less than 400 mm/yr (in the mountainous southern region). Annual PET data were acquired from the publicly accessible WaPOR portal, developed by the Food and Agriculture Organization [80]. The WaPOR portal offers a near real-time database utilizing satellite data, facilitating the monitoring of agricultural water productivity across various spatial scales. The dataset obtained from the WaPOR database comprised multiple resolution levels: continental (250 m ground resolution), country (100 m), and sub-national (30 m). This dataset encompasses several variables, including water productivity, land productivity, actual and potential evapotranspiration, land cover and use, biomass, rainfall, carbon dioxide uptake, yields, harvested index, and crop calendar. For this study, raster data of the annual PET were extracted at the continental level from the year 2009 to the present, with a ground resolution of 250 m, using WaPOR portal data [81]. The raster calculation algorithm was employed to derive the time-averaged annual PET, as depicted in Figure 13. Figure 13 shows that PET varies from a minimum of 2000 mm/yr in the mountainous southern region to a peak of slightly below 3000 mm/yr elsewhere.

Figure 14 exhibits the spatial distribution of the average Aridity Index within the study area. The figure highlights that the majority of the study area falls within the hyper-arid classification, except the mountainous southern and southwestern regions, which are classified as arid zones.

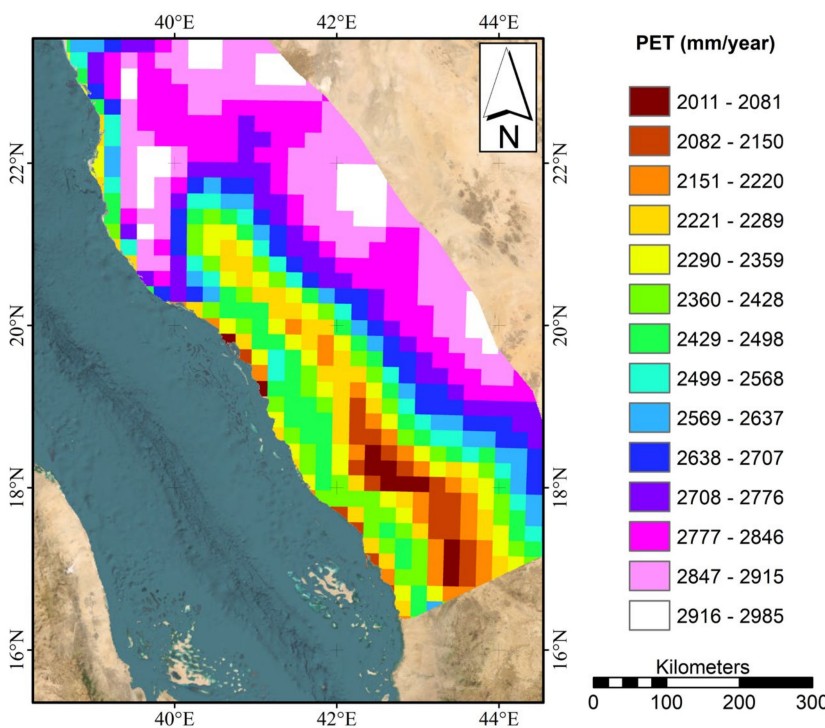

**Figure 13.** Time-averaged annual reference evapotranspiration.

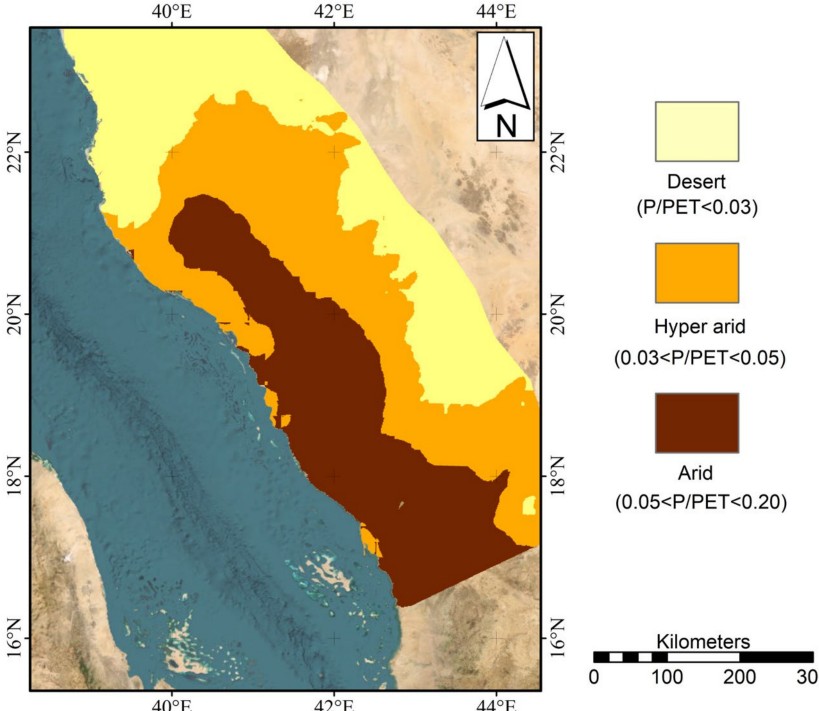

**Figure 14.** Aridity classification in the study area.

## 5. Use Case

This section showcases a detailed example of how to deal with missing rainfall data in a network of stations for specific non-consecutive years (as indicated in Table 7) by using the findings outlined earlier.

**Table 7.** Maximum rainfall records with missing entries.

|  | **J124** | **B001** | **J137** | **J126** | **TA007** | **J127** | **B007** | **B101** |
|---|---|---|---|---|---|---|---|---|
| 2008 | 23.0 | 32.0 | missing | 26.78991 | 26.0 | missing | 46.0 | 19.0 |
| 2009 | 34.0 | 63.0 | missing | 33.0 | 54.5 | missing | 42.5 | 25.0 |
| 2010 | 60.0 | 56.0 | missing | 41.0 | 168.0 | missing | 54.0 | 45.0 |
| 2011 | 28.0 | 82.5 | missing | 22.0 | 60.0 | missing | 44.0 | 27.0 |
| 2012 | 40.0 | missing | missing | 17.0 | 34.5 | missing | 62.0 | 56.0 |
| 2013 | 26.5 | missing | missing | 28.0 | 24.5 | missing | 54.0 | 40.0 |
| 2014 | 7.3 | 30.8 | 70.4 | 35.5 | 37.8 | 40.5 | 36.1 | 41.5 |
| 2015 | 6.5 | 89.0 | 51.3 | 51.5 | missing | 50.8 | 59.0 | missing |
| 2016 | 32.0 | 69.0 | 50.5 | 50.8 | missing | 50.5 | 26.0 | 52.0 |
| 2017 | 1.3 | 78.8 | 24.5 | 21.6 | missing | 45.3 | 26.2 | 15.0 |
| 2018 | 5.0 | 34.4 | 42.5 | 21.5 | missing | 40.5 | 35.8 | 34.5 |

The missing data for the years 2008 to 2010 can be directly retrieved from Figure S5, while the data for the years 2011 to 2013 can be acquired from Figure S6 in the Supplementary Materials, since these years fall after 2014. For the years 2015 to 2018, missing Material can be derived by employing ordinary Kriging interpolation with a J-Bessel semivariogram to generate geostatistical layers using the available data from adjacent stations, as demonstrated in Figure 15. The resulting missing data can be found in Table 8 after the filling process.

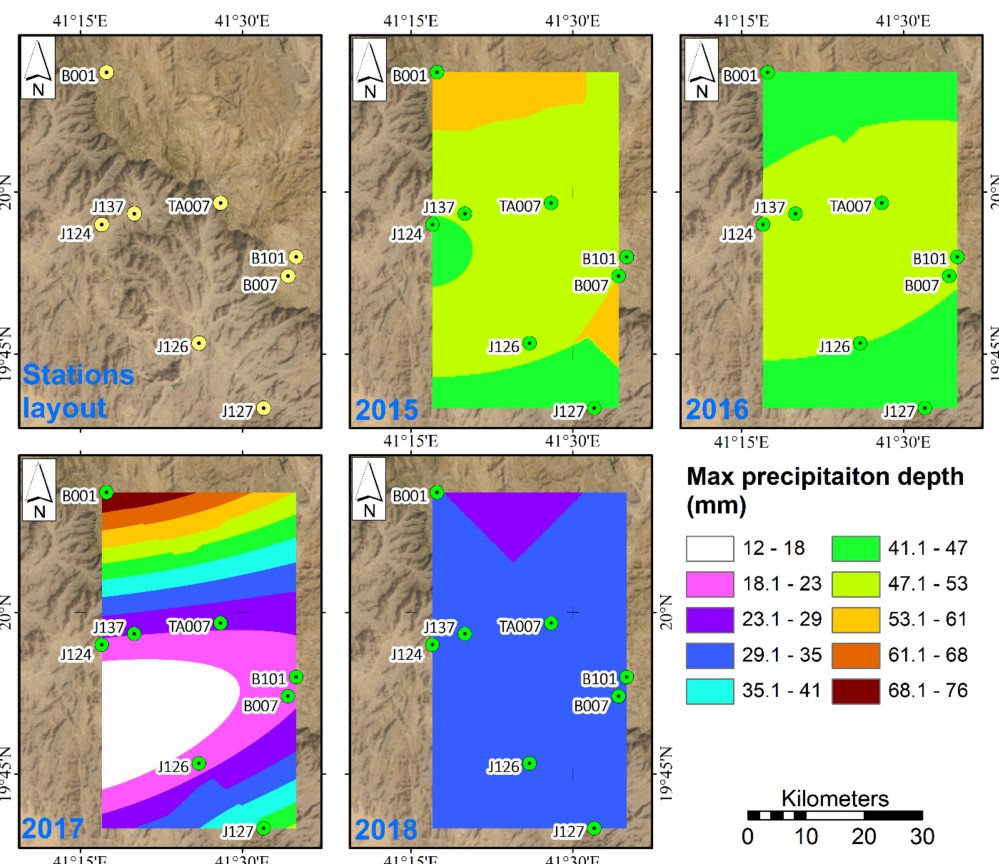

**Figure 15.** Geostatistical surfaces for the years from 2015 to 2018.

**Table 8.** Interpolated filled data.

|      | J124 | B001 | J137 | J126 | TA007 | J127 | B007 | B101 |
|------|------|------|------|------|-------|------|------|------|
| 2008 | 23.0 | 32.0 | 27.2 | 26.8 | 26.0  | 28.2 | 46.0 | 19.0 |
| 2009 | 34.0 | 63.0 | 46.4 | 33.0 | 54.5  | 51.4 | 42.5 | 25.0 |
| 2010 | 60.0 | 56.0 | 46.4 | 41.0 | 168.0 | 42.6 | 54.0 | 45.0 |
| 2011 | 28.0 | 82.5 | 30.4 | 22.0 | 60.0  | 22.9 | 44.0 | 27.0 |
| 2012 | 40.0 | 19.4 | 26.8 | 17.0 | 34.5  | 32.4 | 62.0 | 56.0 |
| 2013 | 26.5 | 30.0 | 30.9 | 28.0 | 24.5  | 32.5 | 54.0 | 40.0 |
| 2014 | 7.3  | 30.8 | 70.4 | 35.5 | 37.8  | 40.5 | 36.1 | 41.5 |
| 2015 | 6.5  | 89.0 | 51.3 | 51.5 | 50.4  | 50.8 | 59.0 | 52.9 |
| 2016 | 32.0 | 69.0 | 50.5 | 50.8 | 47.2  | 50.5 | 26.0 | 52.0 |
| 2017 | 16.0 | 78.8 | 24.5 | 21.6 | 24.4  | 45.3 | 26.2 | 15.0 |
| 2018 | 5.0  | 34.4 | 42.5 | 21.5 | 30.6  | 40.5 | 35.8 | 34.5 |

## 6. Summary and Conclusions

Rainfall depth and spatial distribution play a pivotal role in hydrological and water resources studies. While rain gauges are the conventional and most accurate data source, obtaining adequate records with accurate spatial distributions can be challenging, especially in regions with complex topography and security concerns, such as arid areas. In such contexts, satellite data and geostatistical interpolation techniques can be employed to estimate the spatial distribution between rain gauges and fill data gaps. However, the accuracy of satellite data in our study area was inadequate, underscoring the critical role of geostatistical interpolation in ensuring reliable rainfall spatial distribution.

This study focused on the southwest part of the Kingdom of Saudi Arabia, covering 123 rain gauges and spanning from 1966 to 2013. Throughout the study period, a total of 51 interpolation techniques were evaluated for total and maximum yearly precipitation records. Each year was assigned an optimal interpolation technique for maximum and total yearly precipitation data, determined by cross-validation to find the least centered Root Mean Square Error among the tested techniques. This optimal technique was then utilized to generate maximum and total yearly rainfall spatial distributions over the study area for each year from 1966 to 2013.

Similarly, interpolation techniques were chosen for mean annual values of the gap-filled data for total and maximum yearly rainfall, intended for years beyond 2013. For maximum yearly rainfall interpolation, ordinary Kriging with a J-Bessel semivariogram is recommended, while for total yearly rainfall, simple Kriging with a K-Bessel semivariogram is recommended.

In practical engineering applications within the study area, when utilizing rainfall time series is advised, the rasters provided in the Supplementary Materials can serve as valuable tools to address data gaps from 1966 to 2013. For data gaps extending beyond 2013, the use of KOJ and KSK techniques is recommended for interpolating maximum daily and total yearly rainfall values, respectively.

The average annual interpolated total rainfall depth and average annual potential evapotranspiration were utilized to develop the spatial distribution of the study area's Aridity Index. Consequently, the study area was classified into three zones based on the Aridity Index scale: desert, hyper-arid, and arid zones.

A methodology was introduced, suggesting the incorporation of additional stations to enhance accuracy and reduce cross-validation errors. Intriguingly, the integration of another 23 stations resulted in a 21% improvement in the sum of the cross-validation error for maximum and total yearly precipitation. A practical use case clearly illustrates the proposed methodology, effectively showcasing its utility for engineering purposes.

**Supplementary Materials:** The following supporting information can be downloaded at: https://www.mdpi.com/article/10.3390/su151814028/s1, Figure S1: Distribution of maximum annual rainfall from 1966 to 1974, Figure S2: Distribution of maximum annual rainfall from 1965 to 1983, Figure S3: Distribution of maximum annual rainfall from 1984 to 1992, Figure S4: Distribution of maximum annual rainfall from 1993 to 2001, Figure S5: Distribution of maximum annual rainfall from 2002 to 2010, Figure S6: Distribution of maximum annual rainfall from 2011 to 2013, Figure S7: Distribution of total annual rainfall from 1966 to 1974; Figure S8: Distribution of total annual rainfall from 1965 to 1983, Figure S9: Distribution of total annual rainfall from 1984 to 1992, Figure S10: Distribution of total annual rainfall from 1993 to 2001, Figure S11: Distribution of total annual rainfall from 2002 to 2010, Figure S12: Distribution of total annual rainfall from 2011 to 2013, Figure S13: Total yearly rainfall cross-validation error for (A) existing rainfall stations, (B) with additional proposed stations (I), (C) with additional proposed stations (I) and (II), and 9D) with additional proposed stations (I), (II), and (III), Figure S14: Maximum daily rainfall cross-validation error for (A) existing rainfall stations, (B) with additional proposed stations (I), (C) with additional proposed stations (I) and (II), and (D) with additional proposed stations (I), (II), and (III).

**Author Contributions:** A.M.H.: Conceptualization, Data Collection, Data Analysis, Methodology, Visualization, Writing—Review and Editing. M.E.: Data Analysis, Methodology, Visualization, Writing—Review and Editing, Supervision. M.I.F.: Methodology, Visualization, Writing—Review and Editing. M.S.A.: Data Analysis, Writing—Review and Editing. B.T.E.: Data Collection, Methodology, Visualization, Writing—Original Draft. All authors have read and agreed to the published version of the manuscript.

**Funding:** This work was supported and funded by the Deanship of Scientific Research at Imam Mohammad Ibn Saud Islamic University (IMSIU) (grant number IMSIU-RG23109).

**Institutional Review Board Statement:** Not applicable.

**Informed Consent Statement:** Not applicable.

**Data Availability Statement:** Data will be made available on request.

**Conflicts of Interest:** The authors declare that they have no known competing financial interests or personal relationships that could have appeared to influence the work reported in this paper.

## Appendix A. Spatial Interpolation Equations and Main Characteristics

Spatial interpolation is the estimation of missing measured parameters at specific locations over the study area by utilizing available data. In the current study, two approaches of spatial interpolation were used: (A) deterministic and (B) geostatistical. In the deterministic approach, the relation between points is presented in the form of a mathematical equation. On the other hand, the geostatistical approach is a probabilistic approach considering the spatial variance. Table A1 shows a summary of the advantages and disadvantages of interpolation techniques.

**(A)   Deterministic Approaches**
(A.1.) Inverse Distance Weighting (IDW)

Inverse Distance Weighting is widely used due to its simplicity. Inverse Distance Weighting assigns weights to the surrounding measured data inversely proportional to each point separating distance. The power exponent (P) in the function varies between 0 and 6. With an increase in the power exponent, the weight of each point decreases faster with distance.

$$Z_x = \sum_{i=1}^{N} \lambda_i Z_i \tag{A1}$$

$$\lambda_i = \frac{d_{xi}^{-P}}{\sum_{i=1}^{N} d_{xi}^{-P}} \tag{A2}$$

where:

$Z_x$: the predicted unknown value at point $(x)$.

$\lambda_i$: the weight value of the sampled point $(i)$.
$Z_i$: the value of the sampled point $(i)$.
$d_{xi}$: the distance between the sampled point $(i)$ and the predicted point $(x)$.
$P$: the power of decreasing weight with distance.

(A.2.) Global Polynomial Interpolation

The global polynomial is a multiple regression between all measured data over the study area. It starts from the first order up to the tenth.

$$Z_{(x_i,y_i)} = \beta_o + \beta_1 \cdot x_i + \beta_2 \cdot y_i + \varepsilon(x_i, y_i) \tag{A3}$$

$$Z_{(x_i,y_i)} = \beta_o + \beta_1 \cdot x_i + \beta_2 \cdot y_i + \beta_3 \cdot x_i^2 + \beta_4 \cdot y_i^2 + \beta_5 \cdot x_i \cdot y_i + \beta_6 \cdot x_i^3 + \beta_7 \cdot y_i^3 + \beta_8 \cdot x_i^2 \cdot y_i + \beta_9 \cdot x_i \cdot y_i^2 + \varepsilon(x_i, y_i) \tag{A4}$$

where:

$Z_{(x_i,y_i)}$: location $(x_i, y_i)$ value.
$\varepsilon(x_i, y_i)$: random error.
$\beta$: parameter.

(A.3.) Local Polynomial Interpolation

Local Polynomial Interpolation has the same procedure as Global Polynomial Interpolation. Local Polynomial Interpolation is conducted over individual windows of the study area, not the whole area, as for Global Polynomial Interpolation.

(A.4.) Radial Basis Function

The Radial Basis Function generates an interpolated surface over the whole study area by utilizing one of the following five spline equations.

$$Regularized\ spline\ function — \emptyset(r) = ln\left(\sigma \cdot \frac{r}{2}\right)^2 + E_1(\sigma \cdot r)^2 + C_E \tag{A5}$$

where:

$\emptyset(r)$: Radial Basis Function.
$r$: distance between the predicted point and each data location.
$E_1$: exponential integer [82].
$C_E$: Euler constant [82].

$$Spline\ with\ tension\ function — \emptyset(r) = ln\left(\sigma \cdot \frac{r}{2}\right) + K_o\left(\sigma \cdot \frac{r}{2}\right) + C_E \tag{A6}$$

where:

$K_o$: modified Bessel function

$$Multiquadric\ function — \emptyset(r) = \left(r^2 + \sigma^2\right)^{1/2} \tag{A7}$$

$$Inverse\ multiquadric\ function — \emptyset(r) = \left(r^2 + \sigma^2\right)^{-1/2} \tag{A8}$$

$$Thinplate\ spline\ function — \emptyset(r) = (\sigma \cdot r)^2 ln(\sigma \cdot r) \tag{A9}$$

**(B) Geostatistical Approaches**

The main difference between geostatistical and deterministic interpolation techniques is the consideration of the parameters' spatial variability in the geostatistical approach. The observed variance with distance is transferred to a theoretical one that can be represented mathematically in the interpolation process. Equation (A10) illustrates the calculation of

the semivariance. The variable ($\gamma$) is named semivariance because it calculates half of the variance, not the entire variance.

$$\gamma(h) = \frac{1}{2N(h)} \sum_{i=1}^{N(h)} (Z(u_i) - Z(u_i + h))^2 \tag{A10}$$

where $Z(u_i)$ and $Z(u_i + h)$ are the values of the at the locations $(u_i)$ and $(u_i + h)$, respectively.

Figure A1 shows a schematic diagram of a semivariogram. Several mathematical equations can be utilized to represent the theoretical semivariogram. In the current study, seven mathematical models were evaluated (Circular, Spherical, Exponential, Gaussian, K-Bessel, J-Bessel, and Stable).

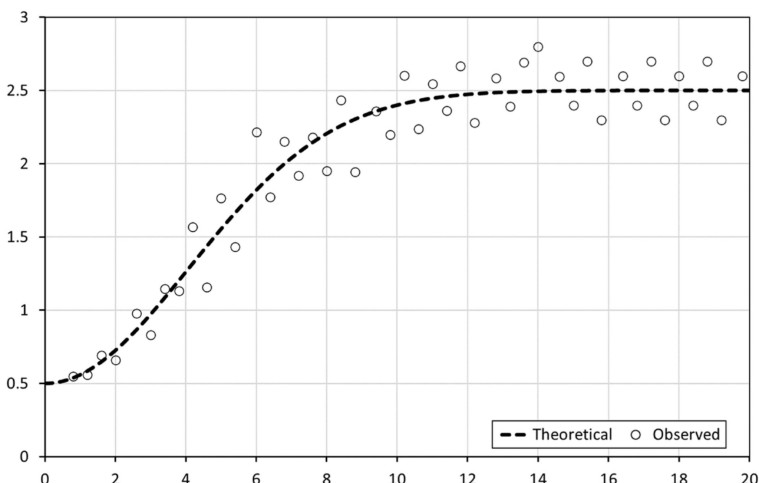

**Figure A1.** Schematic semivariogram.

The geostatistical analysis estimates the missing values at location $(u_o)$ by assigning weights to the measured values, as shown in Equation (A11).

$$z^*(u_o) = \sum_{i=1}^{n} \lambda_i z(u_i) \tag{A11}$$

where:

$z^*(u_o)$ : an estimate of the variable of interest at the location $(u_o)$.
$z(u_i)$ : the measured value of the variable of interest at the location $(u_i)$.
$\lambda_i$ : the Kriging weight of $z(u_i)$.
$n$ : the total number of data locations.

Geostatistical prediction is subjected to two main constraints: (A) the unbiasedness as given in Equation (A12), and (B) minimization of the variance of the estimated value $(z^*(u_o))$, shown in Equation (A13).

$$E[z^*(u_o) - z^*(u_o)] = 0 \Rightarrow \sum_{i=1}^{n} \lambda_i = 1 \tag{A12}$$

$$E[z^*(u_o) - z(u_o)]^2 = \sum_{i=1}^{n} \sum_{j=1}^{n} \lambda_i \lambda_j \gamma_{ij} - 2 \sum_{i=1}^{n} \lambda_i \gamma_{io} + C(o) \tag{A13}$$

(B.1.) Simple Kriging

In simple Kriging, the stationary random parameter mean $(\mu)$ is assumed constant and known before Kriging. The model prediction parameters are given by Equation (A14).

$$
\begin{pmatrix}
\gamma_{11} & \gamma_{12} & \gamma_{13} & & \gamma_{1n} \\
\gamma_{21} & \gamma_{22} & \gamma_{23} & \cdots & \gamma_{2n} \\
\gamma_{31} & \gamma_{32} & \gamma_{33} & & \gamma_{23n} \\
& & \vdots & \ddots & \vdots \\
\gamma_{n1} & \gamma_{n2} & \gamma_{n3} & \cdots & \gamma_{nn}
\end{pmatrix}
\begin{pmatrix}
\lambda_1 \\
\lambda_2 \\
\lambda_3 \\
\vdots \\
\lambda_n
\end{pmatrix}
=
\begin{pmatrix}
\gamma_{01} \\
\gamma_{02} \\
\gamma_{03} \\
\vdots \\
\gamma_{0n}
\end{pmatrix}
\tag{A14}
$$

(B.2.) Ordinary Kriging

In ordinary Kriging, the stationary random parameter mean $(\mu)$ is assumed constant and known before Kriging. The prediction parameters are given by Equation (A15)

$$
\begin{pmatrix}
\gamma_{11} & \gamma_{12} & \gamma_{13} & \cdots & \gamma_{1n} & 1 \\
\gamma_{21} & \gamma_{22} & \gamma_{23} & \cdots & \gamma_{2n} & 1 \\
\gamma_{31} & \gamma_{32} & \gamma_{33} & \cdots & \gamma_{3n} & 1 \\
\vdots & \vdots & \vdots & \ddots & \vdots & 1 \\
\gamma_{n1} & \gamma_{n2} & \gamma_{n3} & \cdots & \gamma_{nn} & 1 \\
1 & 1 & 1 & \cdots & 1 & 0
\end{pmatrix}
\begin{pmatrix}
\lambda_1 \\
\lambda_2 \\
\lambda_3 \\
\\
\lambda_n \\
m
\end{pmatrix}
=
\begin{pmatrix}
\gamma_{01} \\
\gamma_{02} \\
\gamma_{03} \\
\\
\gamma_{0n} \\
1
\end{pmatrix}
\tag{A15}
$$

where:

$m$ : the mean value of the stationary variable.

(B.3.) Universal Kriging

If the concerned parameters exhibit a trend leading to nonstationary mean behavior, the sampling domain can be limited [83]. Kriging with a trend is another name of universal Kriging. In universal Kriging, another set of unbiasedness conditions is required. The prediction parameters are given by Equation (A16).

$$
\begin{pmatrix}
\gamma_{11} & \gamma_{12} & \gamma_{13} & \cdots & \gamma_{1n} & 1 & f_1{}^1 & f_1{}^2 & \cdots & f_1{}^L \\
\gamma_{21} & \gamma_{22} & \gamma_{23} & \cdots & \gamma_{2n} & 1 & f_2{}^1 & f_2{}^2 & \cdots & f_2{}^L \\
\gamma_{31} & \gamma_{32} & \gamma_{33} & \cdots & \gamma_{3n} & 1 & f_3{}^1 & f_3{}^1 & \cdots & f_3{}^L \\
\vdots & \vdots & \vdots & \ddots & \vdots & \vdots & \vdots & \vdots & \ddots & \vdots \\
\gamma_{n1} & \gamma_{n2} & \gamma_{n3} & \cdots & \gamma_{nn} & 1 & f_n{}^1 & f_n{}^2 & \cdots & f_n{}^L \\
1 & 1 & 1 & \cdots & 1 & 0 & 0 & 0 & 0 & 0 \\
f_1{}^1 & f_2{}^1 & f_3{}^1 & \cdots & f_n{}^1 & 0 & 0 & 0 & 0 & 0 \\
f_1{}^2 & f_2{}^2 & f_3{}^2 & \cdots & f_n{}^2 & 0 & 0 & 0 & 0 & 0 \\
\vdots & \vdots & \vdots & \ddots & \vdots & 0 & 0 & 0 & 0 & 0 \\
f_1{}^L & f_2{}^L & f_3{}^L & \cdots & f_n{}^2 & 0 & 0 & 0 & 0 & 0
\end{pmatrix}
\begin{pmatrix}
\lambda_1 \\
\lambda_2 \\
\lambda_3 \\
\vdots \\
\mu_0 \\
\mu_1 \\
\mu_2 \\
\mu_3 \\
\vdots \\
\mu_L
\end{pmatrix}
=
\begin{pmatrix}
\gamma_{01} \\
\gamma_{02} \\
\gamma_{03} \\
\\
\gamma_{0n} \\
f_1{}^0 \\
f_2{}^0 \\
f_3{}^0 \\
\vdots \\
f_L{}^0
\end{pmatrix}
\tag{A16}
$$

where:

$L$: the number of unbiasedness conditions.
$f_P^L$: the $P^{\text{th}}$ basis function.

(B.4.) Cokriging

Cokriging is used when there are several variables. In the current study, we have two variables: (A) the rainfall depth and (B) the elevation of the rain gauges. The mathematical formulas of Cokriging can be found in [84–88].

(B.5.) Empirical Bayesian Kriging

There is a shortage in the literature regarding detailed descriptions of Empirical Bayesian Kriging (EBK). Most of the available documents focus computer package use [68,89].

**Table A1.** Summary of advantages and disadvantages of interpolation techniques.

| Interpolation Technique | Advantages | Disadvantages |
| --- | --- | --- |
| Inverse Distance Weighting | • Simple and easy to apply.<br>• Fast computing time. | • Generates "Bulls Eyes" at measurement locations.<br>• Sensitive to the selection of power factor.<br>• Sensitive to outliers. |
| Global Polynomial Interpolation | • Simple and easy to apply.<br>• Fast computing time.<br>• Provides a smooth interpolated surface over the whole study area. | • Does not capture local trends.<br>• Sensitive to the polynomial degree.<br>• Large influence of the edge points. |
| Local Polynomial Interpolation | • Flexibility (allows the selection between several polynomial degrees).<br>• Captures local trends.<br>• Can handle complex patterns. | • High computational demand.<br>• Sensitive to the polynomial degree.<br>• May generate discontinuity at local region edges |
| Radial Basis Function | • Flexibility (allows the selection between five basis functions).<br>• Captures local trends.<br>• Can capture complex spatial patterns. | • High computational demand.<br>• Sensitive to basis function selection. |
| Kriging | • Provides Best Linear Unbiased Prediction (BLUP).<br>• Several types of Kriging provide flexibility to deal with stationary and non-stationary variables. | • Requires preparing a variogram before starting the computation.<br>• Sensitive to model assumption.<br>• Computationally demanding. |
| Cokriging | • Provide Best Linear Unbiased Prediction (BLUP).<br>• Several types of kriging provide flexibility to deal with stationary and non-stationary variables.<br>• Can handle more than one variable simultaneously.<br>• Incorporation of auxiliary variables can improve prediction accuracy. | • Requires preparing a variogram before starting the computation.<br>• Sensitive to model assumption.<br>• The highest computational demand approach. |
| Empirical Bayesian Kriging | • Automated variogram generation. | • Limited theoretical justificationOutliers sensitivity |

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
