# Peer review of "Evaluation of Geospatial Interpolation Techniques for Enhancing Spatiotemporal Rainfall Distribution and Filling Data Gaps in Asir Region, Saudi Arabia"

_sustainability, doi:10.3390/su151814028_

Round 1

Reviewer 1 Report

I hope this letter finds you well. I appreciate the opportunity to review this work and provide constructive feedback to enhance the quality and clarity of the paper.

Overall, the manuscript presents spatial interpolation techniques and technical study on Rainfall Distribution and Filling Data Gaps.

The objectives of the study were:

1. Assess various spatial interpolation techniques to ascertain the optimal approach for accurate rainfall prediction across diverse arid regions. The authors noted 51 interpolation techniques used in the study. It is suggested to present the techniques with the applicable output in a table with the related method. They had not predicted rainfall so it’s better to remove it.

2. Analyze the sufficiency of rainfall station distribution and pinpoint optimal sites for new rain gauge installation within the study area. The proposed of new gauges should be based on the results of interpolation and identification of areas with high estimation errors. Prior to the proposed map in Figure 11, no relevant explanations are given.

Author Response

We would like to express our sincere gratitude for the time and effort you invested in reviewing our article titled “Evaluation of Geospatial Interpolation Techniques for Enhancing Spatiotemporal Rainfall Distribution and Filling Data Gaps in Asir Region, Saudi Arabia". Your valuable insights and constructive feedback have been instrumental in enhancing the quality and rigor of our work.

We have carefully addressed your comments and suggestions, as shown in the attached file

Reviewer 2 Report

Reviewer report – Manuscript “Evaluation of Geospatial Interpolation Techniques for Enhancing Spatiotemporal Rainfall Distribution and Filling Data Gaps in Asir Region, Saudi Arabia

by Ahmed M. Helmi, Mohamed Elgamal, Mohamed I. Farouk,

Mohamed S. Abdelhamed and Bakinam T. Essawy.

This study focused on filling up data gaps and getting an accurate spatiotemporal distribution of rainfall for the Asir region in southwest Saudi Arabia.

This manuscript is a timely contribution to the field.

Conclusions are presented in an appropriate fashion and are well supported by the data.

This paper, in a major revised form, would be helpful to the community and can be published in this journal. I hope my comments help to guide the authors in that direction.

Genneral comments and suggestions:

1.I believe that showing a table instead of the figure 5 (Codes of the used 51 interpolation techniques in the current study) would be better for the readers, thus more explanations can be added;

2. To accompany sub-chapters 3.2.1 and 3.2.2, an appendix with a description of the methods is required, which should cover interpolation methods in greater detail and include the presentation of the fundamental equations;

3. Table 5 must be completed with 0.80 <= AI < 1.0 Humid, 1.0 <= AI < 2.0 Very humid;

4. The legends of the figures must be uniform in terms of font size, some are much too large.I suggest uniforming the style of the figures.

5. Basic issues like "advantages and disadvantages" for the geospatial interpolation techniques used to arrive at the results, etc., should be highlighted. Meaning of these findings should be added as well to help readers better understand the value of this research;

6. Make a list of abbreviations.

The work should be published since it contains material that the journal's readers will find beneficial, but I think it should be changed in light of past recommendations.

I commend the authors for their hard work and for choosing such a fascinating topic.

Best Wishes,

The reviewer

Author Response

(The authors gave the same response as above.)

Round 2

Reviewer 2 Report

The authors answered my questions and revised the article, the paper can be accepted in present form.

Best wishes,

 The reviewer